# Computation of mixed resolvability for a circular ladder and its unbounded nature

**Sunny Kumar Sharma[1], Vijay Kumar Bhat[1], Muhammad Azeem** [2,3]*,
**Manikonda Gayathri[4], Bandar Almohsen** [5]

1 School of Mathematics, Shri Mata Vaishno Devi University, Katra, Jammu and Kashmir, India,
2 Department of Mathematics, Riphah International University, Lahore, Pakistan, 3 Department of Solids
and Structures, School of Engineering, The University of Manchester, Manchester, United Kingdom,
4 First Grade College (Affiliated to Bangalore North University), Varthur, Bangalore, India, 5 Department
of Mathematics, College of Science, King Saud University, Riyadh, Saudi Arabia

* muhammad.azeem@riphah.edu.pk, azeemali7009@gmail.com

**Peer Review History:** PLOS recognizes the
benefits of transparency in the peer review
process; therefore, we enable the publication of
all of the content of peer review and author
responses alongside final, published articles.
The editorial history of this article is available
here: https://doi.org/10.1371/journal.pone.
0313788

**Data availability statement:** There is no data
available for this manuscript. All the data is
included in this manucript.

**Funding:** The research is supported by
Researchers Supporting Project number
(RSP2025R158), King Saud University, Riyadh,
Saudi Arabia

## Abstract

Let $\Gamma = \Gamma(V,E)$ be a simple, planar, connected, and undirected graph. The article primarily concentrates on a category of planar graphs, detailing the explicit identification of each member within this graph family. Within the domain of graph theory, the parameters used to uniquely identify vertices and edges of a graph are commonly referred to as variants of metric dimension, collectively known as resolvability parameters. The present study focuses on the intricate planar structure of a five-sided circular ladder (pentagonal); denoted by $P_h^5$, and investigate some of the recently introduced resolvability parameters for it, which are mixed metric basis and mixed metric dimension. We prove that the mixed metric dimension for $P_h^5$ is unbounded, and it depends upon the number of vertices present in it. The comparison between several resolvability parameters, viz., metric dimension and edge metric dimension, for $P_h^5$ with mixed metric dimension have also been incorporated in this manuscript, indicating higher level of complexity for resolving both edge and vertex-based relationships. Moreover, several theoretical as well as application based properties, including examples, have also been discussed for $P_h^5$.

## Introduction

Let $\Gamma = (V, E)$ be a non-trivial, simple, connected, and undirected graph, with size $k$ and order $m$. In the last two decades, the concepts of metric dimension (MD) and resolving set (RS) have been the most explored topics in graph theory [1,2]. The concept of MD can be easily understood by comparing it with Global Positioning Systems (GPS) which require a certain number of satellites for their proper functionality in space [3]. The minimum cardinality of the RS, say $F$, is said to be a MD of the graph $\Gamma$, where the members of the set $F$ are called *landmarks or satellites*. The selection of these ordered landmark vertices in $F$ is very important, and this set has some specific property, known as *resolving property*. The property use to uniquely identify each vertex present in $\Gamma$ by means of the distance between the landmark vertices present in $F$ and each vertex of $\Gamma$, is the resolving property [3,4].

**Competing interests:** The authors have declared that no competing interests exist.

Investigating RS and corresponding MD for multiple intricate graph structures is complicated. However, it is worth noting that this activity can provide significant insights that can be leveraged to address a multitude of complex scenarios. As an example, the utilisation of a small set denoted as $F$ has been found to be beneficial in various applications. One such application is aiding robots in their navigation within a physical space. Additionally, this set can also be employed for the identification of intersection points on roads. Furthermore, it has been observed that by assigning a unique identification number (UIN), the group of people can be uniquely specified using this set. Moreover, the set $F$ can also be utilised for tracing the transmission of a disease between different regions [5–7]. This approach can also be applied to various other tasks, such as identifying sources of misinformation and deceptive information within a social network, comparing different network architectures, quantitatively encoding symbolic data, or categorising chemical structures [3].

The notions of RS and MD were first introduced by Slater [2] in 1975, under the terms *locating set* and *location number*, respectively. In the next year, these concepts were again presented by Harary and Melter [1], but they used the terms RS and MD instead of locating set and location number, respectively. While, the concept of the *dimension of a graph*, was already defined in 1965, by Erdos et al. [8]. For this, they have comprehensively described the geometrical interpretation of the dimension of a graph, and carried out an investigation of its distinct implications. The major work in the initial papers [1,2], concentrates on the MD of tree graphs, including other graphs as well, and for that they have obtained RSs and their corresponding MD. In their work, they have discussed the MD for the following graph families; wheel graph, complete graph, cycle graph, and complete bipartite graph. The MD obtained for the wheel graph was reported to be wrong in [1], and later it was corrected by Sooryanarayana et al. [9]. By using the chosen RS and the respective distances between the vertices of RS and the graph's vertices, they have developed an algorithm, that is capable of reconstructing the tree graphs. However, it is important to note that this is not always possible for all varieties of graphs, as not each edge can be guaranteed to be represented by the possible shortest path with the end vertex included in the chosen RS.

The concepts of MD and RS have been advanced by several researchers from both a theoretical and applicative point view. They have also introduced several interesting variants of RSs, for instance, mixed resolving sets, multiresolving sets, partition resolving sets, edge resolving sets, strong resolving sets, fault-tolerant resolving sets, etc. [10–12]. Implementation of these notions in several scientific domains result in a diverse range of applications, such as image processing and pattern recognition [13], network discovery and verification [14], navigation of robots to locate their position in a network [15], the unique identification of the locations of intersecting roads [6], mastermind games involving critical multiple strategies [16], medicinal graph-theoretic chemistry and connected joins in graphs [4], etc. The problem of finding RSs and corresponding MD for any arbitrary graph $\Gamma$ is NP-complete [11,17–19], i.e., for any graph $\Gamma$ and positive natural $k$, it is still computationally difficult to comprehend that $dim_v(\Gamma) \leq k$.

These notions of MD as well as its variants have been investigated for distinctively significant graph families, for instance; planar graphs: path graph, kayak paddle graph, several ladder graphs, (ladders of pentagons, heptagons, nonagons, etc), antiprism graph, mobious ladder graph, wheel graph, cycle graph, various convex polytopes, tadpole graph, and many more; chemical graphs: one-pentagonal, one heptagonal, one nonagonal carbon nanocone structures, linear heptagons structures, polycyclic aromatic compound, and linear phenylene structure, for these one can refer [20–26]. The list is long but is still incomplete, i.e., there are infinite number of distinct graph families for which the notions of RSs and its variants have not been discussed yet. So, to address this partially, in this paper, we consider an interesting

graph family of a planar graph known as five-sided circular ladder, denoted by $P_h^5$, and determine its mixed metric basis as well as mixed metric dimension.

Planar graphs hold significant importance in graph theory due to their rich and practical applications in the real world. The properties of planar graphs, such as the fact that they can be drawn on 2-dimensional plane with no edge crossing, make them highly remarkable in the various scientific disciplines, which include circuit layout optimisation, transportation systems, urban planning, and network design analysis. By utilising the concepts and properties related to planar graphs, engineers and researchers can handle various complex problems effectively, resulting in accurate solutions while taking into account different constraints. The investigation of distinct planar graphs enhanced the knowledge of graph theory, and it helped in solving real-world problems with precision and elegance [27]. Nowadays, the research on planar graphs focuses on the development of highly effective algorithms for investigating their planarity, behavior under distinct graph-theoretic operations, and structural properties. Furthermore, expressing the relationship between several areas of mathematics and planar graphs, for instance, topology, algebraic graph theory, and geometric graph theory, can reveal interesting connections and lay the groundwork for several multidisciplinary applications [6,10]. Next, we list the finding of the manuscript.

## Contribution

The analysis of graphs and networks, with a complete focus on the elements of their vertex sets and edge sets, enables researchers to reveal various vital characteristics of these graphic structures. Dealing with edges and vertices, in this paper, we have considered an interesting family of planar graphs, known as *five-sided circular ladder $P_h^5$*, and studied its mixed metric dimension (MMD). In particular, we will investigate the mixed resolving set for $P_h^5$ and its corresponding MMD. Furthermore, we will compare the existing values of metric dimension and edge metric dimension with the obtained values of MMD for $P_h^5$. Finally, we will also discuss the independence property in the mixed resolving set and find its corresponding independent mixed metric dimension.

## Research methodology

The research methodology used to obtain mixed basis and MMD for $P_h^5$, includes the combination of new results as well as existing results for the comprehensive investigation of the planar graph $P_h^5$. In order to establish a strong foundation to obtain our desired results, we first perform a thorough analysis and review of existing literature regarding planar graphs and resolvability parameters. This process involves a thorough assessment of computational techniques, theoretical frameworks, and previous research on the above-mentioned topics. Then, the new results in this paper are generated by dedicated research initiatives. This includes the formulation of new lemmas, conducting several validity checks, and executing the analysis of data to acquire a broader comprehension of the behaviour and properties of the planar graph $P_h^5$. Next, a comparison between the newly obtained results and existing results would be made to extend and validate the existing knowledge of $P_h^5$. Through a strategic combination of new insights and existing knowledge, this study aims to investigate the planar graph $P_h^5$ thoroughly, with particular focus on its vertices and edges, by employing the above-mentioned research methodology. Then the utilisation of new contributions and pre-existing knowledge, the main objective of this methodology is to comprehend the overall understanding of the specific field of study. It facilitates a holistic approach to advancing our understanding and applications of planar graphs in various domains.

## Structure of the work

This manuscript is arranged in the following manner: Section Preliminaries is devoted to the basics of graph theory, which includes the definitions, notations, and various other graph-theoretic terminologies; also the recent findings regarding $P_h^5$ have been listed in this section. Section Five-sided circular ladder with mixed basis and mixed metric dimension consists of various lemmas and proposition, including the main finding of the manuscript. Section Independent mixed basis and independent mixed metric dimension of $P_h^5$ obtains the independent characteristic on the minimum MRS of $P_h^5$. Section Comparison between different resolvability parameters for $P_h^5$ presents the comparison between metric dimension, edge metric dimension, and MMD for $P_h^5$. Finally, the conclusion and future work of this manuscript is listed in the Section Conclusion.

## Preliminaries

In this section, we undertake a comprehensive examination of several fundamental definitions that are crucial for our study. These definitions include the edge resolving set, resolving set, mixed resolving set, metric basis, edge metric basis, mixed metric basis, and respective dimensions for graphs. Additionally, we incorporate recent research findings pertaining to the graph of $P_h^5$ into this particular section.

Let $\Gamma = (V, E)$ be a simple, connected, non-trivial, undirected planar graph. The distance between two vertices $w$ and $j$ in a graph $\Gamma$ is the length of the shortest path between $w$ and $j$. Let $F = \{p_1, p_2, p_3, \dots, p_r\}$ be an ordered subset of vertices and $x$ be an edge or a vertex in $\Gamma$. Then, the symbol $\xi(x|F) = (d(x, p_1), d(x, p_2), \dots, d(x, p_r))$ is the $r$-tuple code for every element $x$ (vertex/edge) in $\Gamma$. This $r$-tuple is known as the metric code/unique representation (edge metric or mixed metric code, etc) for an element $x$ with respect to the set $F$.

**Definition 2.1. Resolving set and metric dimension:** In $\Gamma$, a set $F_v = \{u_1, u_2, u_3, \dots, u_d\}$ is referred to as a *resolving set*, if for every two distinct vertices $p_1$ and $p_2$ in $\Gamma$, there exists at least one $u_i \in F_v$ such that $d(p_1, u_i) \neq d(p_2, u_i)$. In other words, if $p_1$ and $p_2$ are any two distinct vertices in $\Gamma$, $\xi_v(p_1|F_v) \neq \xi_v(p_2|F_v)$, then $F_v$ is the resolving set in $\Gamma$. The resolving set with least cardinality serves as metric basis in graph $\Gamma$. The number of elements in a metric basis set in $\Gamma$ is called as the *metric dimension* of $\Gamma$ and it is denoted by $dim_v(\Gamma)$ [1,2].

**Definition 2.2. Edge Resolving Set (ERS) and Edge Metric Dimension (EMD):** In $\Gamma$, the distance between an edge and a vertex is defined as follows, $d(e, x) = \min\{d(a, x), d(b, x)\}$, where $e = ab$, $x$, $a$, and $b$ are vertices in $\Gamma$. A set $F_e = \{u_1, u_2, u_3, \dots, u_d\}$ is referred to as an *ERS*, if for every two distinct edges $f_1$ and $f_2$ in $\Gamma$, there exists at least one $u_i \in F_e$ such that $d(f_1, u_i) \neq d(f_2, u_i)$. In other words, if $f_1$ and $f_2$ are any two distinct edges in $\Gamma$ and $\xi_e(f_1|F_e) \neq \xi_e(f_2|F_e)$, then $F_e$ is an ERS in $\Gamma$. An ERS with least cardinality serves as edge metric basis (EMB) in graph $\Gamma$. The number of elements in an edge metric basis set in $\Gamma$ is called as the *edge metric dimension* of $\Gamma$ and it is denoted by $dim_e(\Gamma)$ [11].

**Definition 2.3. Mixed Metric Basis (MMB) and Mixed Metric Dimension (MMD):** In general, mixed metric dimension is the combination of metric dimension and EMD. A set $F_m = \{u_1, u_2, u_3, \dots, u_d\}$ is referred to as an *MMB*, if for every two distinct elements $m_1$ and $m_2$ in $V(\Gamma) \cup E(\Gamma)$, there exists at least one $u_i \in F_m$ such that $d(m_1, u_i) \neq d(m_2, u_i)$. In other words, if $m_1$ and $m_2$ are any two distinct elements (vertex or edge) in $\Gamma$ and $\xi_m(m_1|F_m) \neq \xi_m(m_2|F_m)$, then $F_m$ is an mixed resolving set (MRS) in $\Gamma$. A MRS with least cardinality serves as mixed metric basis (MMB) in $\Gamma$. The number of elements in a mixed metric basis set in $\Gamma$ is called as the *mixed metric dimension* of $\Gamma$ and it is denoted by $mdim_{ve}(\Gamma)$ [28].

If a set $F^i \subseteq V(\Gamma)$ consist of vertices such that no two vertices in it are adjacent, then $F^i$ is said to be an independent set [29]. Now, a set $F_m^i \subseteq V(\Gamma)$ with two properties, viz., i) resolving property and ii) independence property, is said to be an independent resolving set (IRS). The minimum cardinality of IRS is referred to as the independent metric dimension (IMD). Similarly, one can define the concept of independent edge metric dimension (IEMD) and independent mixed metric dimension (IMMD) [30].

## Five-sided circular ladder $P_h^5$

We denote the structure consisting of several five-sided faces, by $P_h^5$, where $h$ represents the number of vertices on the inner-most cycle and 5 is for its five-sided faces. The sets representing the vertices and edges for $P_h^5$, are denoted by $V(P_h^5)$ and $E(P_h^5)$ respectively. These respective sets are defined below.

$$V(P_h^5) \quad = \quad \{s_j, n_j, y_j : 1 \leq j \leq h\}$$

and

$$E(P_h^5) \quad = \quad \{s_j n_j, n_j y_j, s_j s_{j+1}, y_j n_{j+1} : 1 \leq j \leq h\}$$

The planar graph $P_h^5$ has vertices of degree 2 and 3, whereas it has edges of the type 3 – 3 and 3 – 2. More specifically, it has $3h$ number of vertices and $4h$ number of edges, out of $3h$ vertices, $2h$-vertices are of degree 3 and $h$-vertices are of degree 2. Further, it has $2h$-edges are of type 3 – 3 and $2h$-edges are of type 3 – 2. It must be noted that $h \in \mathbb{N}$ and the graph of $P_h^5$ is shown in Fig 1.

We partitioned the set of vertices $V(P_h^5)$ of $P_h^5$ into three parts as follows: $S = \{s_j | 1 \leq j \leq h\}$, $N = \{n_j | 1 \leq j \leq h\}$, and $Y = \{y_j | 1 \leq j \leq h\}$. We call the set $S$ as the set of $s$-vertices, set $N$ as the set of $n$-vertices, and set $Y$ as the set of $y$-vertices. Now, in this manuscript, we consider the graph $P_h^5$ and investigate its mixed basis as well as MMD. For this purpose, we consider $s_1 = s_{h+1}$, $n_1 = n_{h+1}$, and $y_1 = y_{h+1}$. Also, we define three sets, which we call as mixed code sets for $P_h^5$, and are as follows: $S^* = \{\xi_m(s_j | F_m) | 1 \leq j \leq h\}$, $N^* = \{\xi_m(n_j | F_m) | 1 \leq j \leq h\}$, and $Y^* = \{\xi_m(y_j | F_m) | 1 \leq j \leq h\}$.

In 2012, Imran et al. [31] have studied the concept of metric dimension for $P_h^5$ where as in 2023, Sharma and Bhat [32–35] studied the notion of edge metric dimension for $P_h^5$; whenever $h \geq 6$. They have obtained the following result for $P_h^5$ respectively.

**Theorem 2.1.**
$dim_v(P_h^5) = 2$, for each $h \geq 6$.

**Theorem 2.2.**
*For the graph of five-sided circular ladder $P_h^5$ and for each $h \geq 6$,*

$$dim_e(P_h) = \begin{cases} 3, & \text{if } 6 \leq h \leq 14; \\ \lceil \frac{h}{6} \rceil & \text{if } h \geq 15. \end{cases}$$

## Five-sided circular ladder with mixed basis and mixed metric dimension

In order to achieve our desired results, we have the following series of lemmas.

**Lemma 3.1.** *Let $F_m = S \subset V(P_h^5)$, where $S = \{s_j | 1 \leq j \leq h\}$. Then, $F_m$ is not a MRS for $P_h^5$.*

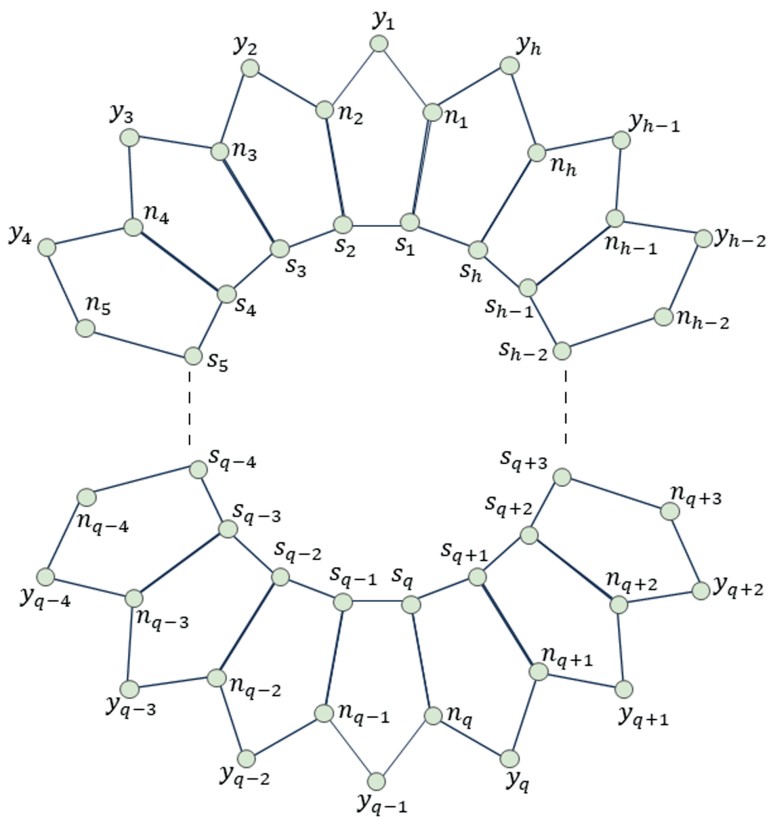

**Fig 1. Planar graph $P_h^5$**

*Proof.* Suppose on the contrary, that $F_m$ is a MRS for $P_h^5$. Then, from Fig 2, we find that the mixed code of an edge $s_1 n_1$ is same as the mixed code of a vertex $s_1$, i.e., $\xi_m(s_1 n_1 | F_m) = \xi_m(s_1 | F_m)$, a contradiction.

Now, we suppose $F_m = S \cup \{n_q\}$, where $n_q \in N = \{n_j | 1 \le j \le h\}$. Then, we have the following lemma.

**Lemma 3.2.** *Let $F_m = S \cup \{n_q\}$ for any q, then $F_m$ is not a MRS for $P_h^5$.*

*Proof.* Suppose on the contrary, that $F_m = S \cup \{n_q\}$, where $1 \le q \le h$, is a MRS for $P_h^5$. Then, from Fig 3, we find that the mixed code of an edge $s_{q+1} n_{q+1}$ is same as the mixed code of a vertex $s_{q+1}$, i.e., $\xi_m(s_{q+1} n_{q+1} | F_m) = \xi_m(s_{q+1} | F_m)$, a contradiction.

Next, we consider the generalization of Lemma 3.2, in the next result.

**Lemma 3.3.** *Let $F_m = S \cup N \setminus \{n_q\}$ for any q, then $F_m$ is not a MRS for $P_h^5$.*

*Proof.* Suppose on the contrary, that $F_m = S \cup N \setminus \{n_q\}$ for any q, where $1 \le q \le h$, is a MRS for $P_h^5$. Then, from Fig 4, we find that the mixed code of an edge $s_q n_q$ is same as the mixed code of a vertex $s_q$, i.e., $\xi_m(s_q n_q | F_m) = \xi_m(s_q | F_m)$, a contradiction.

On investigating the circular ladder from different perspective and observing various types of distances, we have the following interesting lemma.

**Lemma 3.4.** *Let $F_m \subseteq V(P_h^5)$ such that for any $k_1, k_2 \in F_m$, $d(k_1, k_2) \ge 5$, then $F_m$ is not a MRS for $P_h^5$.*

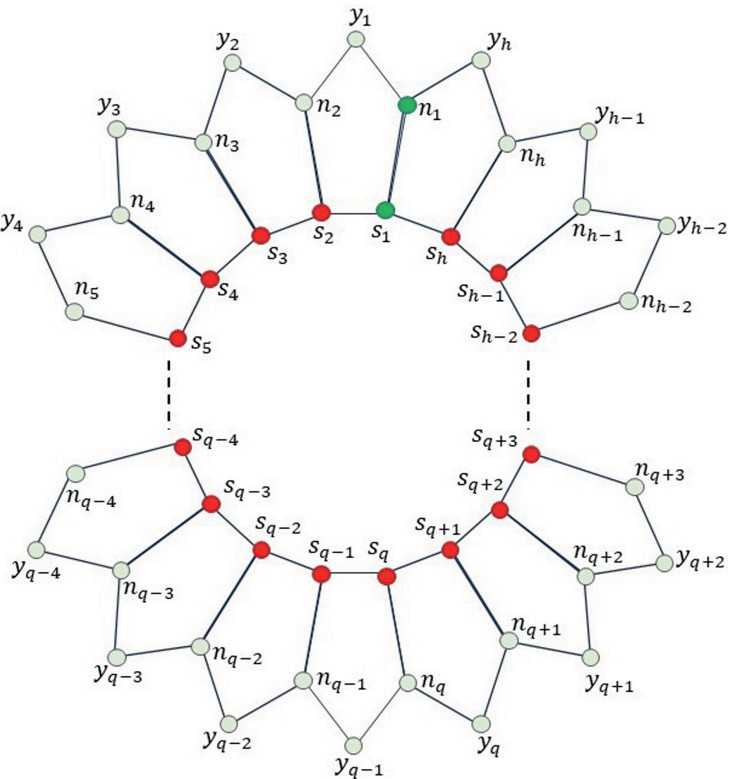

**Fig 2. Pentagonal circular ladder $P_h^5$ for Lemma 3.1.**

*Proof.* Suppose on the contrary, that $F_m$ is a MRS, such that for any $k_1, k_2 \in F_m$, $d(k_1, k_2) \geq 5$. Now, we have following observations.

Case I: Suppose $s_j, s_{j+5} \in F_m$ and so $d(s_j, s_{j+5}) = 5$. Then, from Fig 1, we find that the mixed code of an edge $s_{j+1}n_{j+1}$ is same as the mixed code of a vertex $s_{j+1}$ i.e., $\xi_m(s_{j+1}n_{j+1}|F_m) = \xi_m(s_{j+1}|F_m)$, a contradiction.

Case II: Suppose $n_j, n_{j+3} \in F_m$ and so $d(n_j, n_{j+3}) = 5$. Then, from Fig 1, we find that the mixed code of an edge $s_{j+1}n_{j+1}$ is same as the mixed code of a vertex $s_{j+1}$ i.e., $\xi_m(s_{j+1}n_{j+1}|F_m) = \xi_m(s_{j+1}|F_m)$, a contradiction.

Case III: Suppose $y_j, y_{j+3} \in F_m$ and so $d(y_j, y_{j+3}) \geq 5$. Then, from Fig 1, we find that the mixed code of an edge $s_{j+2}n_{j+2}$ is same as the mixed code of a vertex $s_{j+2}$ i.e., $\xi_m(s_{j+2}n_{j+2}|F_m) = \xi_m(s_{j+2}|F_m)$, a contradiction.

Case IV: Suppose $s_j, n_{j+4} \in F_m$ and so $d(s_j, n_{j+4}) = 5$. Then, from Fig 1, we find that the mixed code of an edge $s_{j+1}n_{j+1}$ is same as the mixed code of a vertex $s_{j+1}$ i.e., $\xi_m(s_{j+1}n_{j+1}|F_m) = \xi_m(s_{j+1}|F_m)$, a contradiction.

Case V: Suppose $s_j, y_{j+3} \in F_m$ and so $d(s_j, y_{j+3}) = 5$. Then, from Fig 1, we find that the mixed code of an edge $s_{j+1}n_{j+1}$ is same as the mixed code of a vertex $s_{j+1}$ i.e., $\xi_m(s_{j+1}n_{j+1}|F_m) = \xi_m(s_{j+1}|F_m)$, a contradiction.

Case VI: Suppose $n_j, y_{j+2} \in F_m$ and so $d(s_j, y_{j+3}) = 5$. Then, from Fig 1, we find that the mixed code of an edge $s_{j+1}n_{j+1}$ is same as the mixed code of a vertex $n_{j+1}$ i.e., $\xi_m(s_{j+1}n_{j+1}|F_m) = \xi_m(n_{j+1}|F_m)$, a contradiction.

Thus, we observed that, if the distance between any two distinct members in $F_m$ is $\geq 5$, then the $F_m$ can never be a MRS for the ladder graph $P_h^5$.

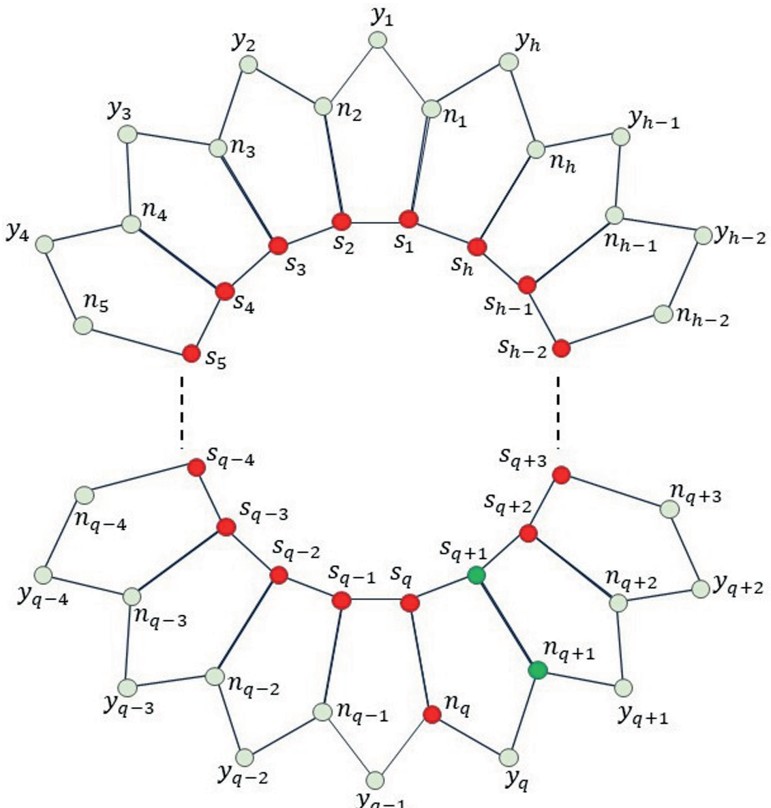

**Fig 3. Pentagonal circular ladder $P_h^5$ for Lemma 3.2.**

From Lemma 3.4, we say that, if $F_m \subseteq V(P_h^5)$ is a MRS for the graph of $P_h^5$, then for any $k_1, k_2 \in F_m$, we have $d(k_1, k_2) \leq 4$. That is, the distance between any two vertices in a MRS $F_m$ of $P_h^5$ is at most 4. Now, our target is to obtain MRS $F_m$ for $P_h^5$ with minimum cardinality. To obtain the minimum MRS for $P_h^5$, we have to select vertices in $F_m$, in such a way that the distance between any two vertices in it is maximum. Based on this fact, we have the following lemma.

**Lemma 3.5.** *Let $F_m \subseteq V(P_h^5)$ be a MRS for $P_h^5$, then $|F_m| \geq \frac{h}{2}$ if $h$ is even and $|F_m| \geq \frac{h+1}{2}$ if $h$ is odd.*

*Proof.* Suppose on the contrary that there exist a MRS $F_m$ with $|F_m| < \frac{h}{2}$ ($h$ is even). Then, there exist at least two elements, say $a$ and $x$, in $V(P_h^5) \cup E(P_h^5)$, such that $d(a, x) \geq 5$, which is a contradiction by Lemma 3.4. Hence, when $h$ is even, then $|F_m| \geq \frac{h}{2}$. Similarly, $|F_m| \geq \frac{h+1}{2}$, whenever $h$ is odd.

Based on the above five lemmas, we find that the MMD of the graph of $P_h^5$ is not bounded as well as constant. Therefore, the MMD of the graph of $P_h^5$ does depend upon the vertices, edges, and several other element of $P_h^5$. Thus, $P_h^5$ has an unbounded and non-constant MMD. Now, we are ready to prove our main result, that the MMD of $P_h^5$ is $\frac{h}{2}$ if $h$ is even natural and $\frac{h+1}{2}$ if $h$ is odd natural. Before, proceeding towards the main result, we have one more interesting lemma, which is given as follows.

**Lemma 3.6.** *Let $F_m = \{n_2, n_4, n_6, n_8, \ldots, n_h\} \subseteq V(P_h^5)$ for even $h$ ($F_m = \{n_2, n_4, n_6, n_8, \ldots, n_{h-1}, n_h\} \subseteq V(P_h^5)$ for odd $h$). Then, $F_m$ is not a MRS for $P_h^5$.*

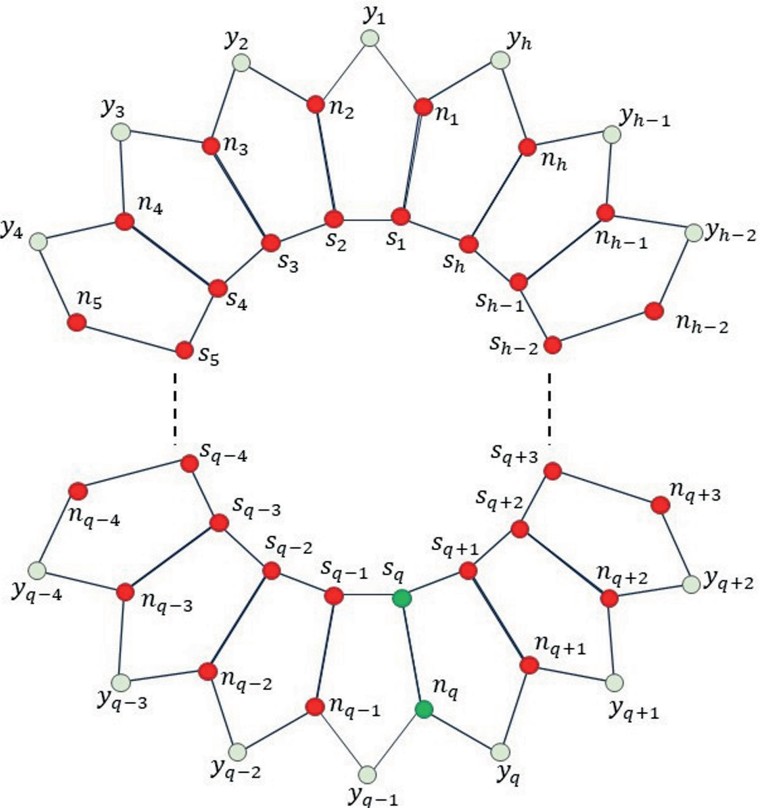

**Fig 4. Pentagonal circular ladder $P_h^5$ for Lemma 3.3.**

*Proof.* Suppose on the contrary, that $F_m = \{n_2, n_4, n_6, n_8, \ldots, n_h\} \subseteq V(P_h^5)$ for even $h$
($F_m = \{n_2, n_4, n_6, n_8, \ldots, n_{h-1}, n_h\} \subseteq V(P_h^5)$ for odd $h$), is a MRS for $P_h^5$. Then, from Fig 5,
we find that the mixed code of an edge $s_1 n_1$ is same as the mixed code of a vertex $s_1$ i.e.,
$\xi_m(s_1 n_1 | F_m) = \xi_m(s_1 | F_m)$, a contradiction.

Now, we have our main result:

**Theorem 3.1.** For the graph of five-sided circular ladder $P_h^5$ and for each $h \geq 6$, we have

$$mdim_{ve}(P_h^5) = \begin{cases} \frac{h}{2} : & \text{if } h \text{ is even;} \\ \frac{h+1}{2} : & \text{if } h \text{ is odd.} \end{cases}$$

*Proof.* The fundamental approach to obtain the desire result, is that, we have to define a
set $F_m$ with a property as follows $F_m$ is the minimum cardinality set with mixed resolving
characteristic for $P_h^5$. Let us define a set $F_m = \{y_2, y_4, y_6, \ldots, y_h\}$ (when $h$ is even and $F_m =$
$\{y_2, y_4, y_6, \ldots, y_{h-3}, y_{h-1}, y_h\}$, when $h$ is odd) for $P_h^5$. We have to prove that the set $F_m$ defined
above is the minimum MRS for $P_h^5$. Now, for some $h$, say $6 \leq h \leq 30$, we have the following
MRSs for $P_h^5$ (shown in Table 1).

From Table 1, we find that for $6 \leq h \leq 31$ the cardinality of the set $F_m$ is $\frac{h}{2}$; when $h$ is even and
the cardinality is $\frac{h+1}{2}$; when $h$ is odd. Next, we have the following Claim.

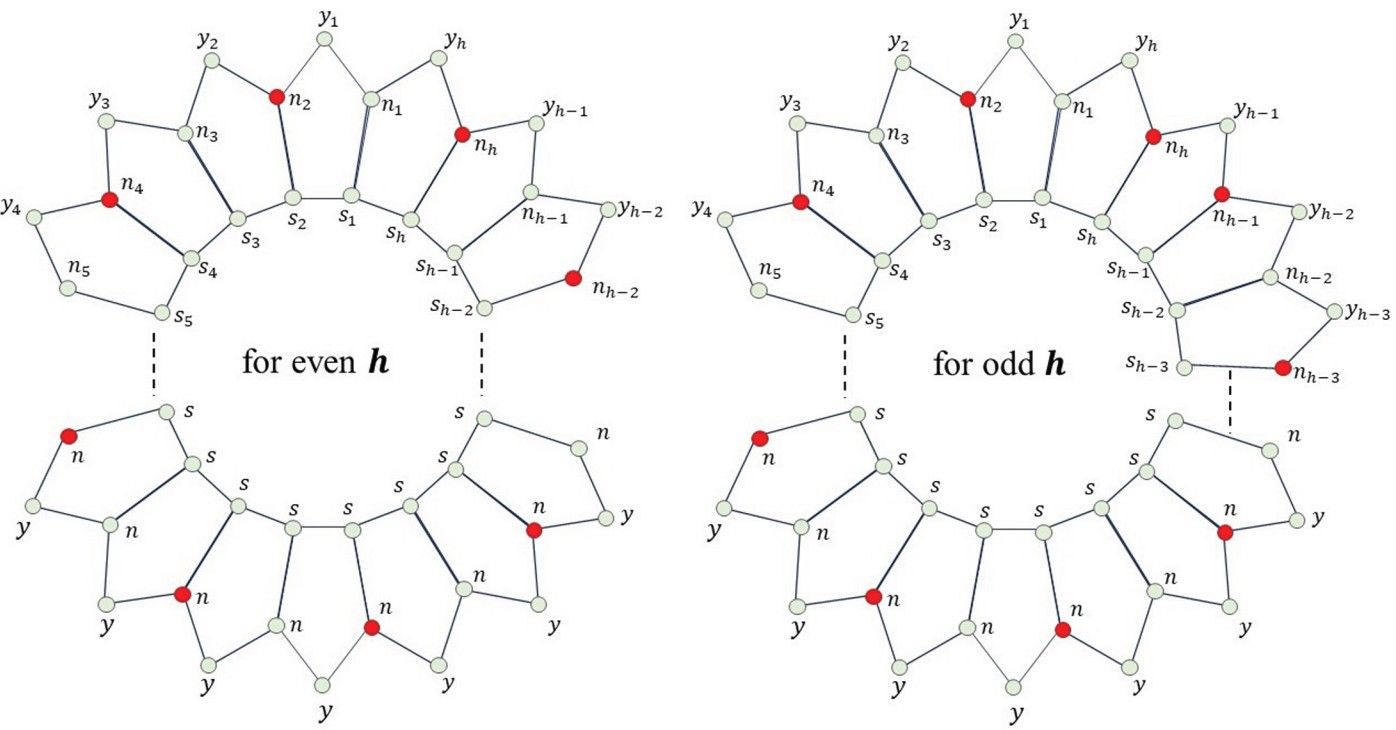

**Fig 5. Pentagonal circular ladder $P_h^5$ for Lemma 3.6 and even and odd $h$.**

**Claim:** The set $F_m = \{y_2, y_4, y_6, \ldots, y_h\}$; when $h$ is even and the set $F_m = \{y_2, y_4, y_6, \ldots, y_{h-3}y_{h-1}, y_h\}$; when $h$ is odd are the minimum MRS for $P_h^5$; for every $h \geq 32$, where $h \in \mathbb{N}$.

In order to prove the claim, we consider two cases based on the values attain by $h$, i.e., i) when $h$ is even and ii) when $h$ is odd.

**Case I** $h \equiv 0 \ (mod\ 2)$ ($h$ is even).

**Table 1. MRSs for $P_h^5$; $6 \leq h \leq 31$.**

| Even $h$ | MRSs $F_m$ | Odd $h$ | MRSs $F_m$ |
|---|---|---|---|
| $h = 6$ | $\{y_j \mid 2 \leq j \leq 6 \ \& \ j \text{ is even}\}$ | $h = 7$ | $\{y_7, y_j \mid 2 \leq j \leq 6 \ \& \ j \text{ is even}\}$ |
| $h = 8$ | $\{y_j \mid 2 \leq j \leq 8 \ \& \ j \text{ is even}\}$ | $h = 9$ | $\{y_9, y_j \mid 2 \leq j \leq 8 \ \& \ j \text{ is even}\}$ |
| $h = 10$ | $\{y_j \mid 2 \leq j \leq 10 \ \& \ j \text{ is even}\}$ | $h = 11$ | $\{y_{11}, y_j \mid 2 \leq j \leq 10 \ \& \ j \text{ is even}\}$ |
| $h = 12$ | $\{y_j \mid 2 \leq j \leq 12 \ \& \ j \text{ is even}\}$ | $h = 13$ | $\{y_{13}, y_j \mid 2 \leq j \leq 12 \ \& \ j \text{ is even}\}$ |
| $h = 14$ | $\{y_j \mid 2 \leq j \leq 14 \ \& \ j \text{ is even}\}$ | $h = 15$ | $\{y_{15}, y_j \mid 2 \leq j \leq 14 \ \& \ j \text{ is even}\}$ |
| $h = 16$ | $\{y_j \mid 2 \leq j \leq 16 \ \& \ j \text{ is even}\}$ | $h = 17$ | $\{y_{17}, y_j \mid 2 \leq j \leq 16 \ \& \ j \text{ is even}\}$ |
| $h = 18$ | $\{y_j \mid 2 \leq j \leq 18 \ \& \ j \text{ is even}\}$ | $h = 19$ | $\{y_{19}, y_j \mid 2 \leq j \leq 18 \ \& \ j \text{ is even}\}$ |
| $h = 20$ | $\{y_j \mid 2 \leq j \leq 20 \ \& \ j \text{ is even}\}$ | $h = 21$ | $\{y_{21}, y_j \mid 2 \leq j \leq 20 \ \& \ j \text{ is even}\}$ |
| $h = 22$ | $\{y_j \mid 2 \leq j \leq 22 \ \& \ j \text{ is even}\}$ | $h = 23$ | $\{y_{23}, y_j \mid 2 \leq j \leq 22 \ \& \ j \text{ is even}\}$ |
| $h = 24$ | $\{y_j \mid 2 \leq j \leq 24 \ \& \ j \text{ is even}\}$ | $h = 25$ | $\{y_{25}, y_j \mid 2 \leq j \leq 24 \ \& \ j \text{ is even}\}$ |
| $h = 26$ | $\{y_j \mid 2 \leq j \leq 26 \ \& \ j \text{ is even}\}$ | $h = 27$ | $\{y_{27}, y_j \mid 2 \leq j \leq 26 \ \& \ j \text{ is even}\}$ |
| $h = 28$ | $\{y_j \mid 2 \leq j \leq 28 \ \& \ j \text{ is even}\}$ | $h = 29$ | $\{y_{29}, y_j \mid 2 \leq j \leq 28 \ \& \ j \text{ is even}\}$ |
| $h = 30$ | $\{y_j \mid 2 \leq j \leq 30 \ \& \ j \text{ is even}\}$ | $h = 31$ | $\{y_{31}, y_j \mid 2 \leq j \leq 30 \ \& \ j \text{ is even}\}$ |

Since $h$ is even, so for more simplification, we write $h = 2q$, where $q \in \mathbb{N}$ and $q \geq 16$. Now, based on these properties of the natural $h$, we consider the set $F_m = \{y_2, y_4, y_6, \ldots, y_h\}$. To complete this case, we have to show that the set $F_m$ is MRS for the graph $P_h^5$. So, instead of the set $F_m$, we consider another set $F_m^*$, which is defined as follows $F_m^* = \{y_2, y_8, y_{2q-4}\}$. Now, we list all the mixed codes corresponding to all the edges as well as vertices present in $P_h^5$ with respect to the set $F_m^*$.

The MCs for the set of edges $SS_e = \{f = s_j s_{j+1}; 1 \leq j \leq h\}$ present in $P_h^5$ are listed in Table 2.

The MCs for the set of edges $SN_e = \{f = s_j n_j; 1 \leq j \leq h\}$ present in $P_h^5$ are listed in Table 3.

The MCs for the set of edges $NY_e = \{f = n_j y_j; 1 \leq j \leq h\}$ present in $P_h^5$ are listed in Table 4.

The MCs for the set of edges $YN_e = \{f = y_j n_{j+1}; 1 \leq j \leq h\}$ present in $P_h^5$ are listed in Table 5.

Now, the sets $SS_e$, $SN_e$, $NY_e$, and $YN_e$ represents the list of MCs for all the edges present in $P_h^5$ with respect to the set $F_m^*$, which consists of only three vertices. Next, we define the MCs for all the vertices present in $P_h^5$.

The MCs for the set of vertices $S = \{v = s_j; 1 \leq j \leq h\}$ present in $P_h^5$ are listed in Table 6.

The MCs for the set of vertices $N = \{v = n_j; 1 \leq j \leq h\}$ present in $P_h^5$ are listed in Table 7.

The MCs for the set of vertices $Y = \{v = y_j; 1 \leq j \leq h\}$ present in $P_h^5$ are listed in Table 8.

From the above list of mixed codes for all the edges and vertices presents in $P_h^5$, we find the same mixed codes for many edges and vertices in $P_h^5$,

1. $\xi_m(n_j|F_m^*) = \xi_m(s_j n_j|F_m^*)$; for $4 \leq j \leq 7$, $10 \leq j \leq 2q - 5$, and $2q - 2 \leq j \leq 2q$ and
2. $\xi_m(n_j y_{j-1}|F_m^*) = \xi_m(n_j y_j|F_m^*)$; for $12 \leq j \leq 2q - 7$.

So, from this fact, we found that the set $F_m^*$ can never be a mixed resolving set for the graph of $P_h^5$, whenever $h$ is even.

**Table 2. MCs for the edges in the set $SS_e$.**

| Edges $f$ | $\xi_m(f|F_m^*)$ |
| --- | --- |
| $s_j s_{j+1}; j = 1$ | $(2, 8, 6)$ |
| $s_j s_{j+1}; j = 2$ | $(2, 7, 7)$ |
| $s_j s_{j+1}; 3 \leq j \leq 7$ | $(j - 1, 9 - j, j + 5)$ |
| $s_j s_{j+1}; j = 8$ | $(7, 2, 13)$ |
| $s_j s_{j+1}; 9 \leq j \leq q - 4$ | $(j - 1, j - 7, j + 5)$ |
| $s_j s_{j+1}; q - 3 \leq j \leq q + 2$ | $(j - 1, j - 7, 2q - j - 3)$ |
| $s_j s_{j+1}; q + 3 \leq j \leq q + 8$ | $(2q - j + 3, j - 7, 2q - j - 3)$ |
| $s_j s_{j+1}; q + 9 \leq j \leq 2q - 5$ | $(2q - j + 3, 2q - j + 9, 2q - j - 3)$ |
| $s_j s_{j+1}; j = 2q - 4$ | $(2q - j + 3, 2q - j + 9, 2)$ |
| $s_j s_{j+1}; 2q - 3 \leq j \leq 2q$ | $(2q - j + 3, 2q - j + 9, j - 2q + 5)$ |

**Table 3. MCs for the edges in the set $SN_e$.**

| Edges $f$ | $\xi_m(f|F_m^*)$ |
| --- | --- |
| $s_j n_j; j = 1$ | $(3, 9, 6)$ |
| $s_j n_j; 2 \leq j \leq 3$ | $(1, 10 - j, j + 5)$ |
| $s_j n_j; 4 \leq j \leq 7$ | $(j - 1, 10 - j, j + 5)$ |
| $s_j n_j; 8 \leq j \leq 9$ | $(j - 1, 1, j + 5)$ |
| $s_j n_j; 10 \leq j \leq q - 4$ | $(j - 1, j - 7, j + 5)$ |
| $s_j n_j; q - 3 \leq j \leq q + 2$ | $(j - 1, j - 7, 2q - j - 2)$ |
| $s_j n_j; q + 3 \leq j \leq q + 8$ | $(2q - j + 4, j - 7, 2q - j - 2)$ |
| $s_j n_j; q + 9 \leq j \leq 2q - 5$ | $(2q - j + 4, 2q - j + 10, 2q - j - 2)$ |
| $s_j n_j; 2q - 4 \leq j \leq 2q - 3$ | $(2q - j + 4, 2q - j + 10, 1)$ |
| $s_j n_j; 2q - 2 \leq j \leq 2q$ | $(2q - j + 4, 2q - j + 10, j - 2q + 5)$ |

**Table 4. MCs for the edges in the set $NY_e$.**

| Edges $f$ | $\xi_m(f|F_m^*)$ | Edges $f$ | $\xi_m(f|F_m^*)$ |
|---|---|---|---|
| $n_j y_j; j = 1$ | $(2, 10, 7)$ | $n_j y_j; 12 \leq j \leq q - 4$ | $(j, j - 6, j + 6)$ |
| $n_j y_j; j = 2$ | $(0, 9, 8)$ | $n_j y_j; q - 3 \leq j \leq q + 2$ | $(j, j - 6, 2q - j - 1)$ |
| $n_j y_j; j = 3$ | $(1, 8, 9)$ | $n_j y_j; q + 3 \leq j \leq q + 8$ | $(2q - j + 5, j - 6, 2q - j - 1)$ |
| $n_j y_j; j = 4$ | $(3, 7, 10)$ | $n_j y_j; q + 9 \leq j \leq 2q - 7$ | $(2q - j + 5, 2q - j + 11, 2q - j - 1)$ |
| $n_j y_j; j = 5$ | $(5, 6, 11)$ | $n_j y_j; j = 2q - 6$ | $(2q - j + 5, 2q - j + 11, 4)$ |
| $n_j y_j; j = 6$ | $(6, 4, 12)$ | $n_j y_j; j = 2q - 5$ | $(2q - j + 5, 2q - j + 11, 2)$ |
| $n_j y_j; j = 7$ | $(7, 2, 13)$ | $n_j y_j; j = 2q - 4$ | $(2q - j + 5, 2q - j + 11, 0)$ |
| $n_j y_j; j = 8$ | $(8, 0, 14)$ | $n_j y_j; j = 2q - 3$ | $(2q - j + 5, 2q - j + 11, 1)$ |
| $n_j y_j; j = 9$ | $(9, 1, 15)$ | $n_j y_j; j = 2q - 2$ | $(2q - j + 5, 2q - j + 11, 3)$ |
| $n_j y_j; j = 10$ | $(10, 3, 16)$ | $n_j y_j; j = 2q - 1$ | $(2q - j + 5, 2q - j + 11, 5)$ |
| $n_j y_j; j = 11$ | $(11, 5, 17)$ | $n_j y_j; j = 2q$ | $(4, 2q - j + 11, 6)$ |

**Table 5. MCs for the edges in the set $YN_e$.**

| Edges $f$ | $\xi_m(f|F_m^*)$ | Edges $f$ | $\xi_m(f|F_m^*)$ |
|---|---|---|---|
| $y_l n_{l+1}; l = 1$ | $(1, 9, 8)$ | $y_l n_{l+1}; 11 \leq l \leq q - 5$ | $(l + 1, l - 5, l + 7)$ |
| $y_j n_{j+1}; j = 2$ | $(0, 8, 9)$ | $y_j n_{j+1}; q - 4 \leq j \leq q + 1$ | $(j + 1, j - 5, 2q - j - 2)$ |
| $y_j n_{j+1}; j = 3$ | $(2, 7, 10)$ | $y_j n_{j+1}; q + 2 \leq j \leq q + 7$ | $(2q - j + 4, j - 5, 2q - j - 2)$ |
| $y_j n_{j+1}; j = 4$ | $(4, 6, 11)$ | $y_j n_{j+1}; q + 8 \leq j \leq 2q - 7$ | $(2q - j + 4, 2q - j + 10, 2q - j - 2)$ |
| $y_j n_{j+1}; j = 5$ | $(6, 5, 12)$ | $y_j n_{j+1}; j = 2q - 6$ | $(2q - j + 4, 2q - j + 10, 3)$ |
| $y_j n_{j+1}; j = 6$ | $(7, 3, 13)$ | $y_j n_{j+1}; j = 2q - 5$ | $(2q - j + 4, 2q - j + 10, 1)$ |
| $y_j n_{j+1}; j = 7$ | $(8, 1, 14)$ | $y_j n_{j+1}; j = 2q - 4$ | $(2q - j + 4, 2q - j + 10, 0)$ |
| $y_j n_{j+1}; j = 8$ | $(9, 0, 15)$ | $y_j n_{j+1}; j = 2q - 3$ | $(2q - j + 4, 2q - j + 10, 2)$ |
| $y_j n_{j+1}; j = 9$ | $(10, 2, 16)$ | $y_j n_{j+1}; j = 2q - 2$ | $(2q - j + 4, 2q - j + 10, 4)$ |
| $y_j n_{j+1}; j = 10$ | $(11, 4, 17)$ | $y_j n_{j+1}; j = 2q - 1$ | $(2q - j + 4, 2q - j + 10, 6)$ |
| | | $y_j n_{j+1}; j = 2q$ | $(3, 2q - j + 10, 7)$ |

**Table 6. MCs for the vertices in the set $S$.**

| Vertices $v$ | $\xi_m(v|F_m^*)$ |
|---|---|
| $s_j; j = 1$ | $(3, 9, 6)$ |
| $s_j; j = 2$ | $(2, 8, 7)$ |
| $s_j; 3 \leq j \leq 8$ | $(j - 1, 9 - j, j + 5)$ |
| $s_j; 9 \leq j \leq q - 4$ | $(j - 1, j - 7, j + 5)$ |
| $s_j; q - 3 \leq j \leq q + 2$ | $(j - 1, j - 7, 2q - j - 2)$ |
| $s_j; q + 3 \leq j \leq q + 8$ | $(2q - j + 4, j - 7, 2q - j - 2)$ |
| $s_j; q + 9 \leq j \leq 2q - 4$ | $(2q - j + 4, 2q - j + 10, 2q - j - 2)$ |
| $s_j; 2q - 3 \leq j \leq 2q$ | $(2q - j + 4, 2q - j + 10, j - 2q + 5)$ |

Now, on considering the following set of vertices, say $F_m = F_m^* \cup \{y_4, y_6, y_{10}, \ldots, y_{2q-6}, y_{2q-2}, y_{2q}\}$ (red color vertices depicted in Fig 6) and again applying the same definition of MRS on all the vertices and edges of $P_h^5$, we find that $\xi_m(a|F_m) \neq \xi_m(x|F_m)$; for all $a, x \in V(P_h^5) \cup E(P_h^5)$. This proves that the new set $F_m$ consisting of $\frac{h}{2}$ vertices from $P_h^5$ is the mixed resolving set with minimum cardinality, i.e., $F_m \setminus \{b\}$ can never be a MRS for $P_h^5$ for any element $b \in V(P_h^5) \cup E(P_h^5)$. This means that $|F_m| \leq \frac{h}{2}$. Therefore, from this fact and Lemma 3.5, we find that $mdim_{ve}(P_h^5) = \frac{h}{2}$ in this particular case for even $h$.

**Case II:** $h \equiv 1 \ (mod \ 2)$ ($h$ is odd).

Since $h$ is odd, so for more simplification, we write $h = 2q + 1$, where $q \in \mathbb{N}$ and $q \geq 16$. Now, based on these properties of the natural $h$, we consider the set $F_m = \{y_2, y_4, y_6, \ldots, y_{h-3}, y_{h-1}, y_h\}$. To complete this case, we have to show that the set $F_m$ is MRS for the graph $P_h^5$. So, instead of the set $F_m$, we consider another set $F_m^*$, which is defined as follows $F_m^* = \{y_2, y_8, y_{2q-4}\}$. Now,

**Table 7. MCs for the vertices in the set $N$.**

| Vertices $v$ | $\xi_m(v\|F_m^*)$ | Vertices $v$ | $\xi_m(v\|F_m^*)$ |
|---|---|---|---|
| $n_j; j = 1$ | $(3, 10, 7)$ | $n_j; q - 3 \leq j \leq q + 2$ | $(j, j - 6, 2q - j - 1)$ |
| $n_j; j = 2$ | $(1, 9, 8)$ | $n_j; q + 3 \leq j \leq q + 8$ | $(2q - j + 5, j - 6, 2q - j - 1)$ |
| $n_j; j = 3$ | $(1, 8, 9)$ | $n_j; q + 9 \leq j \leq 2q - 6$ | $(2q - j + 5, 2q - j + 11, 2q - j - 1)$ |
| $n_j; j = 4$ | $(3, 7, 10)$ | $n_j; j = 2q - 5$ | $(2q - j + 5, 2q - j + 11, 3)$ |
| $n_j; 5 \leq j \leq 6$ | $(j, 11 - j, j + 6)$ | $n_j; j = 2q - 4$ | $(2q - j + 5, 2q - j + 11, 1)$ |
| $n_j; j = 7$ | $(j, 3, j + 6)$ | $n_j; j = 2q - 3$ | $(2q - j + 5, 2q - j + 11, 1)$ |
| $n_j; j = 8$ | $(j, 1, j + 6)$ | $n_j; j = 2q - 2$ | $(2q - j + 5, 2q - j + 11, 3)$ |
| $n_j; j = 9$ | $(j, 1, j + 6)$ | $n_j; j = 2q - 1$ | $(2q - j + 5, 2q - j + 11, j - 2q + 6)$ |
| $n_j; j = 10$ | $(j, 3, j + 6)$ | $n_j; j = 2q$ | $(2q - j + 5, 2q - j + 11, j - 2q + 6)$ |
| $n_j; 11 \leq j \leq q-$ | $(j, j - 6, j + 6)$ | | |

**Table 8. MCs for the vertices in the set $Y$.**

| Vertices $v$ | $\xi_m(v\|F_m^*)$ | Vertices $v$ | $\xi_m(v\|F_m^*)$ |
|---|---|---|---|
| $y_j; j = 1$ | $(2, 10, 8)$ | $y_j; 11 \leq j \leq q - 4$ | $(j + 1, j - 5, j + 7)$ |
| $y_j; j = 2$ | $(0, 9, 9)$ | $y_j; q - 3 \leq j \leq q + 2$ | $(j + 1, j - 5, 2q - j - 1)$ |
| $y_j; j = 3$ | $(2, 8, 10)$ | $y_j; q + 3 \leq j \leq q + 8$ | $(2q - j + 5, j - 5, 2q - j - 1)$ |
| $y_j; j = 4$ | $(4, 7, 11)$ | $y_j; q + 9 \leq j \leq 2q - 7$ | $(2q - j + 5, 2q - j + 11, 2q - j - 1)$ |
| $y_j; j = 5$ | $(6, 6, 12)$ | $y_j; j = 2q - 6$ | $(2q - j + 5, 2q - j + 11, 4)$ |
| $y_j; j = 6$ | $(7, 4, 13)$ | $y_j; j = 2q - 5$ | $(2q - j + 5, 2q - j + 11, 2)$ |
| $y_j; j = 7$ | $(8, 2, 14)$ | $y_j; j = 2q - 4$ | $(2q - j + 5, 2q - j + 11, 0)$ |
| $y_j; j = 8$ | $(9, 0, 15)$ | $y_j; j = 2q - 3$ | $(2q - j + 5, 2q - j + 11, 2)$ |
| $y_j; j = 9$ | $(10, 2, 16)$ | $y_j; j = 2q - 2$ | $(2q - j + 5, 2q - j + 11, 4)$ |
| $y_j; j = 10$ | $(11, 4, 17)$ | $y_j; 2q - 1 \leq j \leq 2q$ | $(2q - j + 5, 2q - j + 11, j - 2q + 7)$ |

we list all the mixed codes corresponding to all the edges as well as vertices present in $P_h^5$ with respect to the set $F_m^*$.

The MCs for the set of edges $SS_e = \{f = s_j s_{j+1}; 1 \leq j \leq h\}$ present in $P_h^5$ are listed in Table 9.
The MCs for the set of edges $SN_e = \{f = s_j n_j; 1 \leq j \leq h\}$ present in $P_h^5$ are listed in Table 10.
The MCs for the set of edges $NY_e = \{f = n_j y_j; 1 \leq j \leq h\}$ present in $P_h^5$ are listed in Table 11.
The MCs for the set of edges $YN_e = \{f = y_j n_{j+1}; 1 \leq j \leq h\}$ present in $P_h^5$ are listed in Table 12.
Now, the sets $SS_e$, $SN_e$, $NY_e$, and $YN_e$ represents the list of MCs for all the edges present in $P_h^5$ with respect to the set $F_m^*$, which consists of only three vertices. Next, we define the MCs for all the vertices present in $P_h^5$.
The MCs for the set of vertices $S = \{v = s_j; 1 \leq j \leq h\}$ present in $P_h^5$ are listed in Table 13.
The MCs for the set of vertices $N = \{v = n_j; 1 \leq j \leq h\}$ present in $P_h^5$ are listed in Table 14.
The MCs for the set of vertices $Y = \{v = y_j; 1 \leq j \leq h\}$ present in $P_h^5$ are listed in Table 15.
From the above list of mixed codes for all the edges and vertices presents in $P_h^5$, we find the same mixed codes for many edges and vertices in $P_h^5$:

1. $\xi_m(n_j|F_m^*) = \xi_m(s_j n_j|F_m^*)$; for $j = 1$, $4 \leq j \leq 7$, $10 \leq j \leq 2q - 5$, and $2q - 2 \leq j \leq 2q + 1$ &
2. $\xi_m(n_j y_{j-1}|F_m^*) = \xi_m(n_j y_j|F_m^*)$; for $12 \leq j \leq 2q - 7$.

So, from this fact, we found that the set $F_m^*$ can never be a mixed resolving set for the graph of $P_h^5$, whenever $h$ is odd.
Now, on considering the following set of vertices, say $F_m = F_m^* \cup \{y_4, y_6, y_{10}, \ldots, y_{2q-6}, y_{2q-2}, y_{2q}, y_{2q+1}\}$ (red color vertices depicted in Fig 7) and again applying the same definition of MRS on all the vertices and edges of $P_h^5$, we find that $\xi_m(a|F_m) \neq \xi_m(x|F_m)$; for all $a, x \in V(P_h^5) \cup E(P_h^5)$.

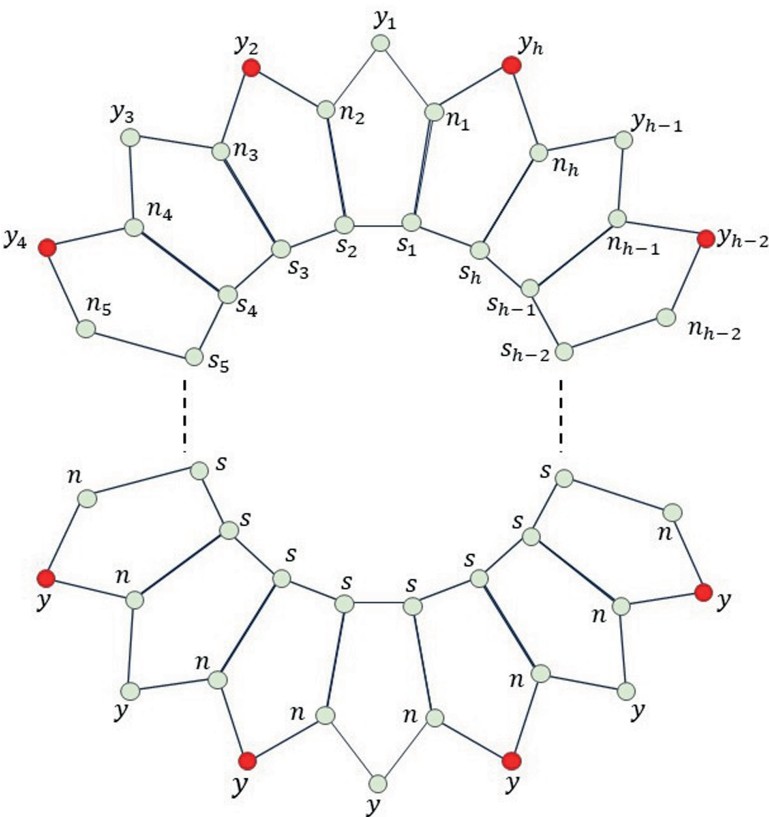

**Fig 6. Pentagonal circular ladder $P_h^5$ for even $h$.**

**Table 9. MCs for the edges in the set $SS_e$.**

| Edges $f$ | $\xi_m(f|F_m^*)$ |
|---|---|
| $s_j s_{j+1}; j = 1$ | $(2, 8, 7)$ |
| $s_j s_{j+1}; j = 2$ | $(2, 7, 8)$ |
| $s_j s_{j+1}; 3 \leq j \leq 7$ | $(j - 1, 9 - j, j + 6)$ |
| $s_j s_{j+1}; j = 8$ | $(7, 2, 14)$ |
| $s_j s_{j+1}; 9 \leq j \leq q - 5$ | $(j - 1, j - 7, j + 6)$ |
| $s_j s_{j+1}; q - 4 \leq j \leq q + 2$ | $(j - 1, j - 7, 2q - j - 3)$ |
| $s_j s_{j+1}; q + 3 \leq j \leq q + 8$ | $(2q - j + 4, j - 7, 2q - j - 3)$ |
| $s_j s_{j+1}; q + 9 \leq j \leq 2q - 5$ | $(2q - j + 4, 2q - j + 10, 2q - j - 3)$ |
| $s_j s_{j+1}; j = 2q - 4$ | $(2q - j + 4, 2q - j + 10, 2)$ |
| $s_j s_{j+1}; 2q - 3 \leq j \leq 2q + 1$ | $(2q - j + 4, 2q - j + 10, j - 2q + 5)$ |

This proves that the new set $F_m$ consisting of $\frac{h+1}{2}$ vertices from $P_h^5$ is the mixed resolving set with minimum cardinality, i.e., $F_m \setminus \{b\}$ can never be a MRS for $P_h^5$ for any element $b \in V(P_h^5) \cup E(P_h^5)$. This means that $|F_m| \leq \frac{h+1}{2}$. Therefore, from this fact and Lemma 3.5, we find that $mdim_{ve}(P_h^5) = \frac{h+1}{2}$ in this particular case for odd $h$.

**Table 10. MCs for the edges in the set $SN_e$.**

| Edges $f$ | $\xi_m(f|F_m^*)$ |
|---|---|
| $s_j n_j; j = 1$ | $(3, 9, 7)$ |
| $s_j n_j; j = 2$ | $(1, 8, 8)$ |
| $s_j n_j; j = 3$ | $(1, 7, 9)$ |
| $s_j n_j; 4 \leq j \leq 7$ | $(j-1, 10-j, j+6)$ |
| $s_j n_j; 8 \leq j \leq 9$ | $(j-1, 1, j+6)$ |
| $s_j n_j; 10 \leq j \leq q-4$ | $(j-1, j-7, j+6)$ |
| $s_j n_j; q-3 \leq j \leq q+3$ | $(j-1, j-7, 2q-j-2)$ |
| $s_j n_j; q+4 \leq j \leq q+9$ | $(2q-j+5, j-7, 2q-j-2)$ |
| $s_j n_j; q+10 \leq j \leq 2q-5$ | $(2q-j+5, 2q-j+11, 2q-j-2)$ |
| $s_j n_j; 2q-4 \leq j \leq 2q-3$ | $(2q-j+5, 2q-j+11, 1)$ |
| $s_j n_j; 2q-2 \leq j \leq 2q+1$ | $(2q-j+5, 2q-j+11, j-2q+5)$ |

**Table 11. MCs for the edges in the set $NY_e$.**

| Edges $f$ | $\xi_m(f|F_m^*)$ | Edges $f$ | $\xi_m(f|F_m^*)$ |
|---|---|---|---|
| $n_j y_j; j = 1$ | $(2, 10, 8)$ | $n_j y_j; 12 \leq j \leq q-4$ | $(j, j-6, j+7)$ |
| $n_j y_j; j = 2$ | $(0, 9, 9)$ | $n_j y_j; q-3 \leq j \leq q+2$ | $(j, j-6, 2q-j-1)$ |
| $n_j y_j; j = 3$ | $(1, 8, 10)$ | $n_j y_j; q+3 \leq j \leq q+8$ | $(2q-j+5, j-6, 2q-j-1)$ |
| $n_j y_j; j = 4$ | $(3, 7, 11)$ | $n_j y_j; q+9 \leq j \leq 2q-7$ | $(2q-j+5, 2q-j+11, 2q-j-1)$ |
| $n_j y_j; j = 5$ | $(5, 6, 12)$ | $n_j y_j; j = 2q-6$ | $(2q-j+5, 2q-j+11, 4)$ |
| $n_j y_j; j = 6$ | $(6, 4, 13)$ | $n_j y_j; j = 2q-5$ | $(2q-j+5, 2q-j+11, 2)$ |
| $n_j y_j; j = 7$ | $(7, 2, 14)$ | $n_j y_j; j = 2q-4$ | $(2q-j+5, 2q-j+11, 0)$ |
| $n_j y_j; j = 8$ | $(8, 0, 15)$ | $n_j y_j; j = 2q-3$ | $(2q-j+5, 2q-j+11, 1)$ |
| $n_j y_j; j = 9$ | $(9, 1, 16)$ | $n_j y_j; j = 2q-2$ | $(2q-j+5, 2q-j+11, 3)$ |
| $n_j y_j; j = 10$ | $(10, 3, 17)$ | $n_j y_j; 2q-1 \leq j \leq 2q+1$ | $(2q-j+5, 2q-j+11, j-2q+6)$ |
| $n_j y_j; j = 11$ | $(11, 5, 18)$ | | |

**Table 12. MCs for the edges in the set $YN_e$.**

| Edges $f$ | $\xi_m(f|F_m^*)$ | Edges $f$ | $\xi_m(f|F_m^*)$ |
|---|---|---|---|
| $y_j n_{j+1}; j = 1$ | $(1, 9, 9)$ | $y_j n_{j+1}; 11 \leq j \leq q-5$ | $(j+1, j-5, j+7)$ |
| $y_j n_{j+1}; j = 2$ | $(0, 8, 10)$ | $y_j n_{j+1}; q-4 \leq j \leq q+1$ | $(j+1, j-5, 2q-j-2)$ |
| $y_j n_{j+1}; j = 3$ | $(2, 7, 11)$ | $y_j n_{j+1}; q+2 \leq j \leq q+8$ | $(2q-j+5, j-5, 2q-j-2)$ |
| $y_j n_{j+1}; j = 4$ | $(4, 6, 12)$ | $y_j n_{j+1}; q+9 \leq j \leq 2q-7$ | $(2q-j+5, 2q-j+11, 2q-j-2)$ |
| $y_j n_{j+1}; j = 5$ | $(6, 5, 13)$ | $y_j n_{j+1}; j = 2q-6$ | $(2q-j+5, 2q-j+11, 3)$ |
| $y_j n_{j+1}; j = 6$ | $(7, 3, 14)$ | $y_j n_{j+1}; j = 2q-5$ | $(2q-j+5, 2q-j+11, 1)$ |
| $y_j n_{j+1}; j = 7$ | $(8, 1, 15)$ | $y_j n_{j+1}; j = 2q-4$ | $(2q-j+5, 2q-j+11, 0)$ |
| $y_j n_{j+1}; j = 8$ | $(9, 0, 16)$ | $y_j n_{j+1}; j = 2q-3$ | $(2q-j+5, 2q-j+11, 2)$ |
| $y_j n_{j+1}; j = 9$ | $(10, 2, 17)$ | $y_j n_{j+1}; j = 2q-2$ | $(2q-j+5, 2q-j+11, 4)$ |
| $y_j n_{j+1}; j = 10$ | $(11, 4, 18)$ | $y_j n_{j+1}; 2q-1 \leq j \leq 2q$ | $(2q-j+5, 2q-j+11, j-2q+7)$ |
| | | $y_j n_{j+1}; j = 2q+1$ | $(3, 2q-j+11, j-2q+7)$ |

## Independent mixed basis and independent mixed metric dimension of $P_h^5$

Now, in this part, we consider the planar graph $P_h^5$, and investigates its structure for independent mixed basis with minimum cardinality. From previous theorem, we find that the cardinality of minimum mixed resolving set depends upon the number of vertices presents in set

**Table 13. MCs for the vertices in the set $S$.**

| Vertices $v$ | $\xi_m(v\vert F_m^*)$ |
|---|---|
| $s_j; j = 1$ | $(3, 9, 7)$ |
| $s_j; j = 2$ | $(2, 8, 8)$ |
| $s_j; 3 \leq j \leq 8$ | $(j - 1, 10 - j, j + 6)$ |
| $s_j; 9 \leq j \leq q - 4$ | $(j - 1, j - 7, j + 6)$ |
| $s_j; q - 3 \leq j \leq q + 3$ | $(j - 1, j - 7, 2q - j - 2)$ |
| $s_j; q + 4 \leq j \leq q + 9$ | $(2q - j + 5, j - 7, 2q - j - 2)$ |
| $s_j; q + 10 \leq j \leq 2q - 4$ | $(2q - j + 5, 2q - j + 11, 2q - j - 2)$ |
| $s_j; 2q - 3 \leq j \leq 2q + 1$ | $(2q - j + 5, 2q - j + 11, j - 2q + 5)$ |

**Table 14. MCs for the vertices in the set $N$.**

| Vertices $v$ | $\xi_m(v\vert F_m^*)$ | Vertices $v$ | $\xi_m(v\vert F_m^*)$ |
|---|---|---|---|
| $n_j; j = 1$ | $(3, 10, 8)$ | $n_j; j = 11$ | $(11, 5, 18)$ |
| $n_j; j = 2$ | $(1, 9, 9)$ | $n_j; 12 \leq j \leq q - 4$ | $(j, j - 6, j + 6)$ |
| $n_j; j = 3$ | $(1, 8, 10)$ | $n_j; q - 3 \leq j \leq q + 4$ | $(j, j - 6, 2q - j - 1)$ |
| $n_j; j = 4$ | $(3, 7, 11)$ | $n_j; q + 5 \leq j \leq q + 9$ | $(2q - j + 6, j - 6, 2q - j - 1)$ |
| $n_j; j = 5$ | $(5, 6, 12)$ | $n_j; q + 10 \leq j \leq 2q - 6$ | $(2q - j + 6, 2q - j + 12, 2q - j - 1)$ |
| $n_j; j = 6$ | $(6, 5, 13)$ | $n_j; j = 2q - 5$ | $(2q - j + 6, 2q - j + 12, 3)$ |
| $n_j; j = 7$ | $(7, 3, 14)$ | $n_j; 2q - 4 \leq j \leq 2q - 3$ | $(2q - j + 6, 2q - j + 12, 1)$ |
| $n_j; j = 8$ | $(8, 1, 15)$ | $n_j; j = 2q - 2$ | $(2q - j + 6, 2q - j + 12, 3)$ |
| $n_j; j = 9$ | $(9, 1, 16)$ | $n_j; 2q - 1 \leq j \leq 2q + 1$ | $(2q - j + 6, 2q - j + 12, j - 2q + 6)$ |
| $n_j; j = 10$ | $(10, 3, 17)$ | | |

**Table 15. MCs for the vertices in the set $Y$.**

| Vertices $v$ | $\xi_m(v\vert F_m^*)$ | Vertices $v$ | $\xi_m(v\vert F_m^*)$ |
|---|---|---|---|
| $y_j; j = 1$ | $(2, 10, 9)$ | $y_j; 11 \leq j \leq q - 4$ | $(j + 1, j - 5, j + 7)$ |
| $y_j; j = 2$ | $(0, 9, 10)$ | $y_j; q - 3 \leq j \leq q + 2$ | $(j + 1, j - 5, 2q - j - 1)$ |
| $y_j; j = 3$ | $(2, 8, 11)$ | $y_j; q + 3 \leq j \leq q + 8$ | $(2q - j + 6, j - 5, 2q - j - 1)$ |
| $y_j; j = 4$ | $(4, 7, 12)$ | $y_j; q + 9 \leq j \leq 2q - 7$ | $(2q - j + 6, 2q - j + 12, 2q - j - 1)$ |
| $y_j; j = 5$ | $(6, 6, 13)$ | $y_j; j = 2q - 6$ | $(2q - j + 6, 2q - j + 12, 4)$ |
| $y_j; j = 6$ | $(7, 4, 14)$ | $y_j; j = 2q - 5$ | $(2q - j + 6, 2q - j + 12, 2)$ |
| $y_j; j = 7$ | $(8, 2, 15)$ | $y_j; j = 2q - 4$ | $(2q - j + 6, 2q - j + 12, 0)$ |
| $y_j; j = 8$ | $(9, 0, 16)$ | $y_j; j = 2q - 3$ | $(2q - j + 6, 2q - j + 12, 2)$ |
| $y_j; j = 9$ | $(10, 2, 17)$ | $y_j; j = 2q - 2$ | $(2q - j + 6, 2q - j + 12, 4)$ |
| $y_j; j = 10$ | $(11, 4, 14)$ | $y_j; 2q - 1 \leq j \leq 2q$ | $(2q - j + 6, 2q - j + 12, j - 2q + 7)$ |
| | | $y_j; j = 2q + 1$ | $(4, 2q - j + 12, j - 2q + 7)$ |

$Y = \{y_j \vert 1 \leq j \leq h\}$. Also, we note that all the vertices present in MRS for $P_h^5$, are not adjacent to one and other. So, considering both of these facts about $P_h^5$, we have the following result.

**Theorem 4.** For the graph of five-sided circular ladder $P_h^5$ and for each $h \geq 6$, the IMMD is as follows

$$mdim_{ve}^i(P_h^5) = \begin{cases} \frac{h}{2} : & \text{if } h \text{ is even;} \\ \frac{h+1}{2} : & \text{if } h \text{ is odd.} \end{cases}$$

*Proof.* The proof is similar to the proof of Theorem 3.1.

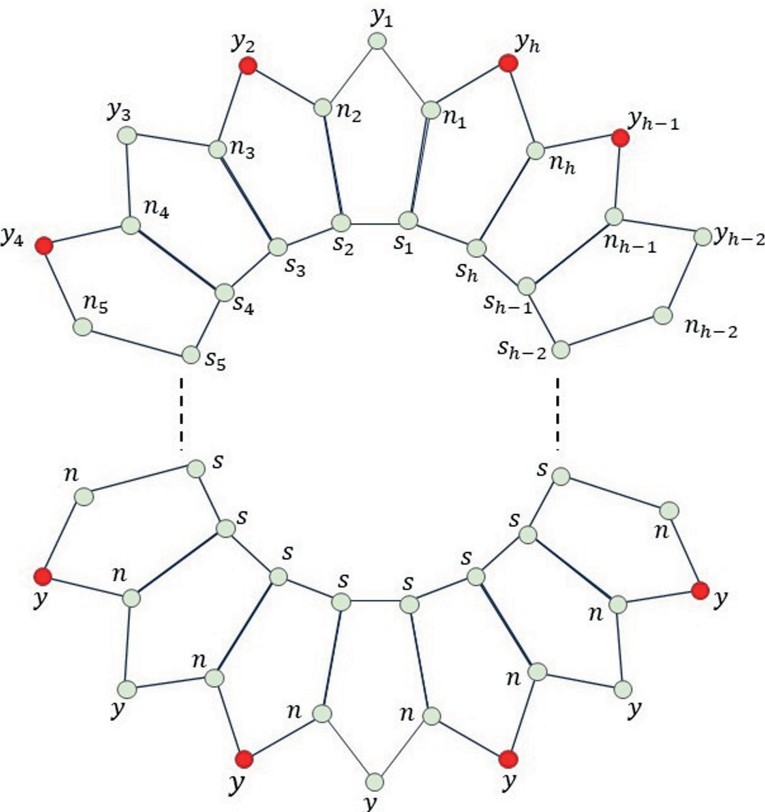

**Fig 7. Pentagonal circular ladder $P_h^5$ for odd $h$.**

## Comparison between different resolvability parameters for $P_h^5$

The results obtained for the graph of $P_h^5$ presents an interesting relation between MD, EMD, MMD, and NV (where NV = number of vertices on the inner cycle of $P_h^5$). We observed the difference, whenever $h$ is even and when it is odd. Although, for every $h \geq 7$, we have obtained the following relation; $dim_v(P_h^5) < dim_e(P_h^5) < mdim_{ve}(P_h^5)$ and when $h = 6$, we have $dim_v(P_6^5) < dim_e(P_6^5) = mdim_{ve}(P_6^5)$ (by Theorem 2.1 and 2.2 (preliminary section), and by Theorem 3.1 (Section Five-sided circular ladder with mixed basis and mixed metric dimension). The Fig 8, shows the difference between MD, EMD, and MMD of $P_h^5$ for even $h$ and Fig 9, shows the difference between MD, EMD, and MMD of $P_h^5$ for odd $h$.

## Example

In this part, we consider an example of $P_h^5$, where $h = 8$, as shown in Fig 10. For this, we investigate its mixed resolving set and which is given as follows:

$F_m = \{y_2, y_4, y_6, y_8, y_{10}, y_{12}, y_{14}, y_{16}\}$ (by Theorem 3.1 and the vertices in $F_m$ are clearly shown in Fig 10 in red color). Now, we present mixed codes for $P_8^5$ in the following two

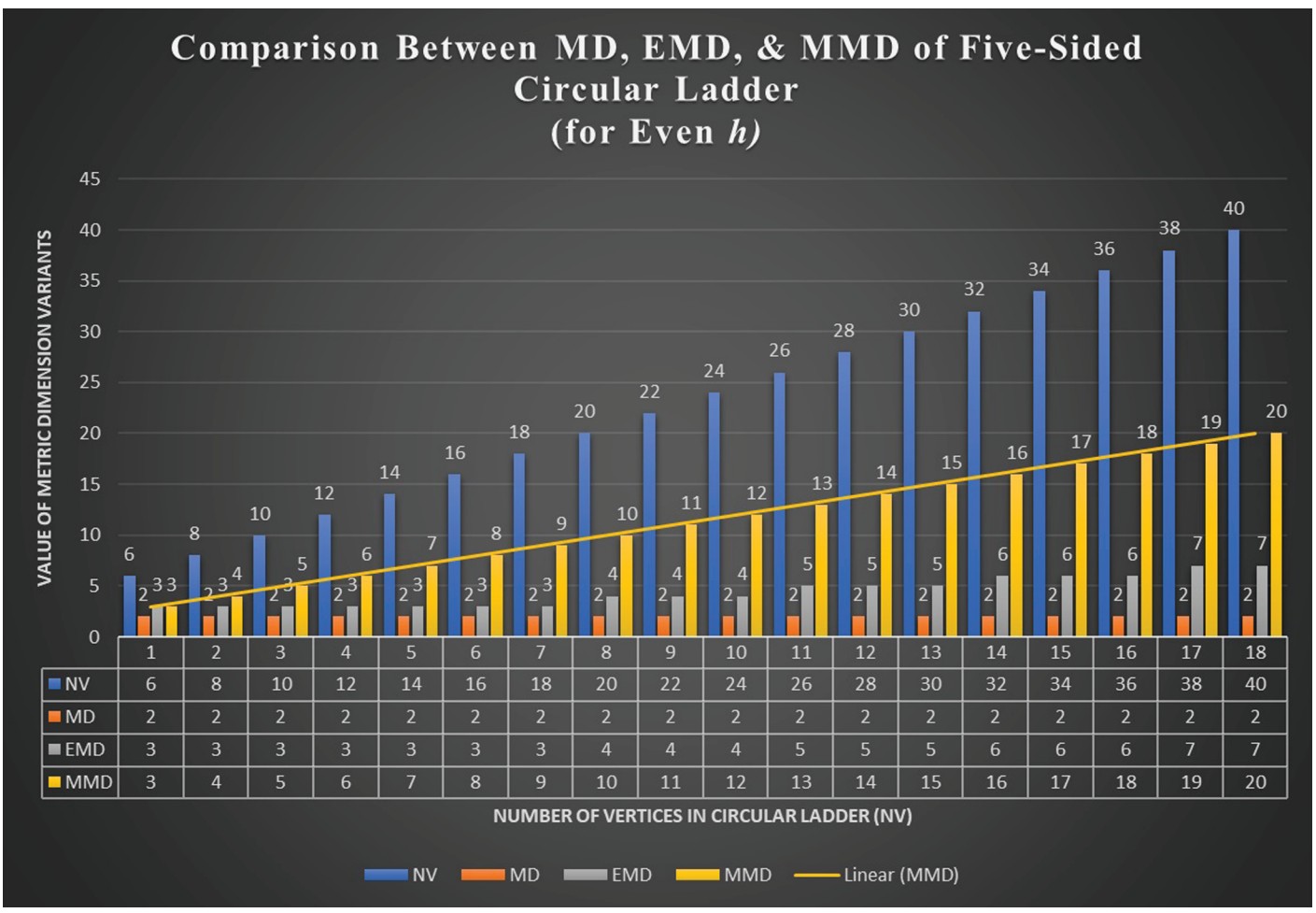

**Fig 8. Comparison between MD, EMD, and MMD of $P_h^5$; for even $h$.**

tables below, i.e., Table 16 (represent mixed codes for all vertices present in $P_h^5$ with respect to $F_m$) and Table 17 (represent mixed codes for all edges present in $P_h^5$ with respect to $F_m$).

## Conclusion

This study is focused on the computation of the MMD of an interesting planar graph family $P_h^5$. For the planar graph $P_h^5$, we found that the MMD is non-constant as well as unbounded, and in particular, we say that the MMD of $P_h^5$ depends on the vertices present in it. Additionally, we have also compared the values of the metric and edge metric dimension of $P_h^5$ with the values for MMD. After, comparing these values, we have observed that $mdim_v e(P_h^5) > dim_e(P_h^5) > dim_v(P_h^5)$. From these observations, one can say that the nature and values of the

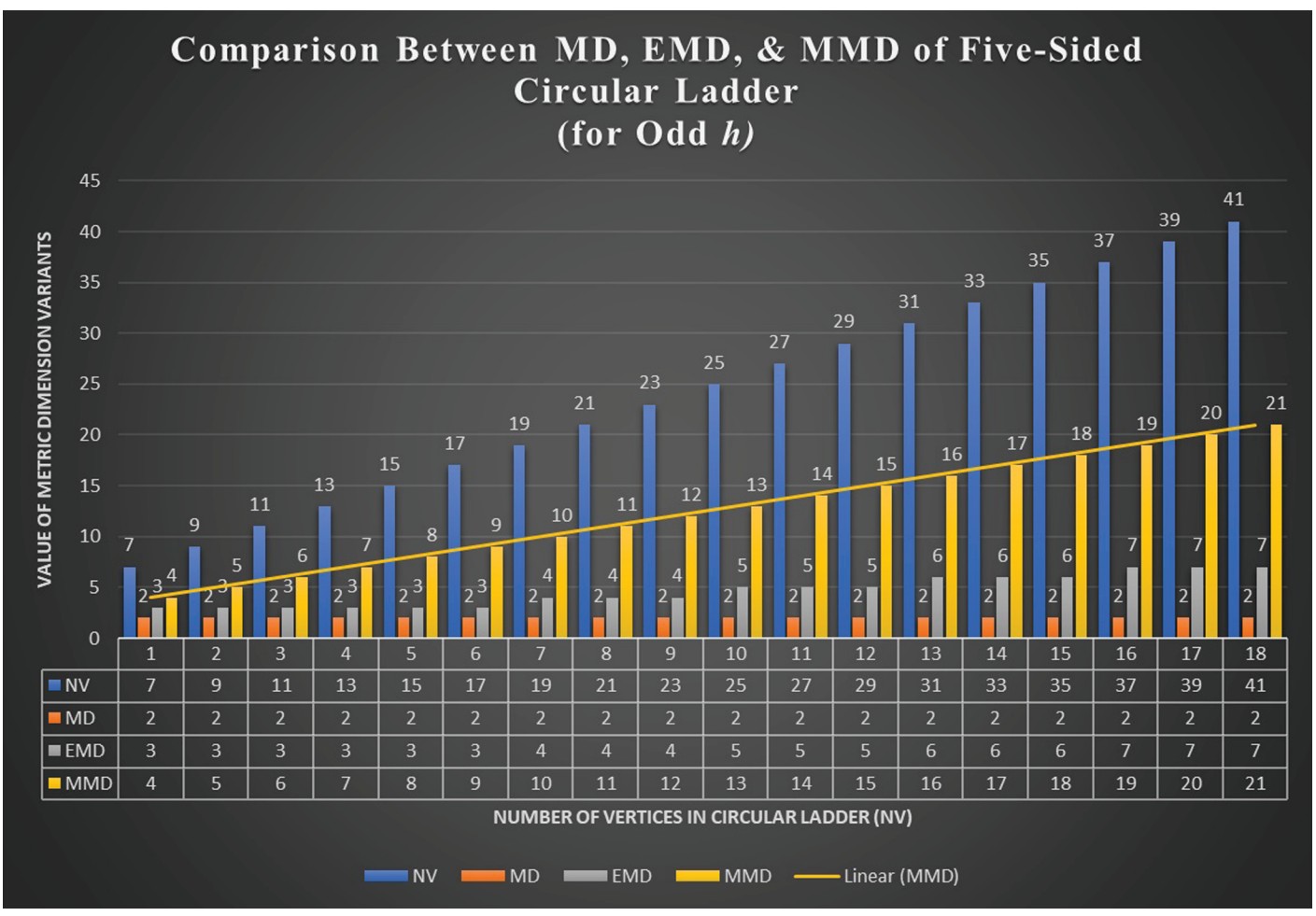

**Fig 9. Comparison between MD, EMD, and MMD of $P_h^5$; for odd $h$.**

MMD of $P_h^5$ are increasing more rapidly than the values of the metric and edge metric dimensions. By giving comparative insights and the values for MMD of $P_h^5$, the research presented here makes a valuable contribution to the broader realm of graph theory. This research may offers a significant insights for applications in various scientific domains, including, optimisation, decision-making processes, and various network designs. The future scope of this research paper directs us in the exploration of new planar graph families, including the investigation of several variants of metric dimension for them.

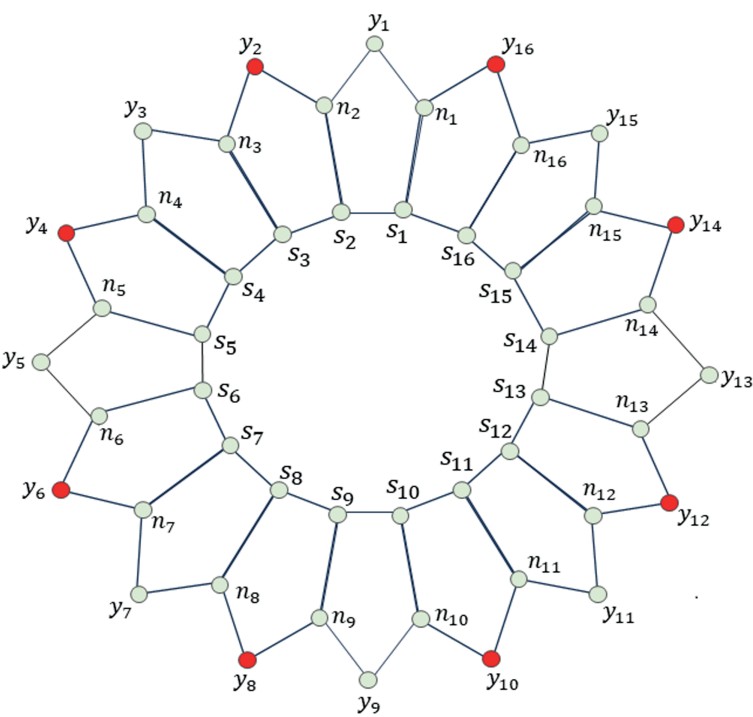

**Fig 10. $P_h^5$; when $h = 8$.**

**Table 16. MCs for all the vertices in $P_8^5$.**

| Vertices $v$ | $\xi_m(v\|F_m)$ | Vertices $v$ | $\xi_m(v\|F_m)$ | Vertices $v$ | $\xi_m(v\|F_m)$ |
|---|---|---|---|---|---|
| $s_1$ | $(3,5,7,9,8,6,4,2)$ | $n_1$ | $(3,6,8,10,9,7,5,1)$ | $y_1$ | $(2,6,8,10,10,8,6,2)$ |
| $s_2$ | $(2,4,6,8,9,7,5,3)$ | $n_2$ | $(1,5,7,9,10,8,6,3)$ | $y_2$ | $(0,4,7,9,11,9,7,4)$ |
| $s_3$ | $(2,3,5,7,9,8,6,4)$ | $n_3$ | $(1,3,6,8,10,9,7,5)$ | $y_3$ | $(2,2,6,8,10,10,8,6)$ |
| $s_4$ | $(3,2,4,6,8,9,7,5)$ | $n_4$ | $(3,1,5,7,9,10,8,6)$ | $y_4$ | $(4,0,4,7,9,11,9,7)$ |
| $s_5$ | $(4,2,3,5,7,9,8,6)$ | $n_5$ | $(5,1,3,6,8,10,9,7)$ | $y_5$ | $(6,2,2,6,8,10,10,8)$ |
| $s_6$ | $(5,3,2,4,6,8,9,7)$ | $n_6$ | $(6,3,1,5,7,9,10,8)$ | $y_6$ | $(7,4,0,4,7,9,11,9)$ |
| $s_7$ | $(6,4,2,3,5,7,9,8)$ | $n_7$ | $(7,5,1,3,6,8,10,9)$ | $y_7$ | $(8,6,2,2,6,8,10,10)$ |
| $s_8$ | $(7,5,3,2,4,6,8,9)$ | $n_8$ | $(8,6,3,1,5,7,9,10)$ | $y_8$ | $(9,7,4,0,4,7,9,11)$ |
| $s_9$ | $(8,6,4,2,3,5,7,9)$ | $n_9$ | $(9,7,5,1,3,6,8,10)$ | $y_9$ | $(10,8,6,2,2,6,8,10)$ |
| $s_{10}$ | $(9,7,5,3,2,4,6,8)$ | $n_{10}$ | $(10,8,6,3,1,5,7,9)$ | $y_{10}$ | $(11,9,7,4,0,4,7,9)$ |
| $s_{11}$ | $(9,8,6,4,2,3,5,7)$ | $n_{11}$ | $(10,9,7,5,1,3,6,8)$ | $y_{11}$ | $(10,10,8,6,2,2,6,8)$ |
| $s_{12}$ | $(8,9,7,5,3,2,4,6)$ | $n_{12}$ | $(9,10,8,6,3,1,5,7)$ | $y_{12}$ | $(9,11,9,7,4,0,4,7)$ |
| $s_{13}$ | $(7,9,8,6,4,2,3,5)$ | $n_{13}$ | $(8,10,9,7,5,1,3,6)$ | $y_{13}$ | $(8,10,10,8,6,2,2,6)$ |
| $s_{14}$ | $(6,8,9,7,5,3,2,4)$ | $n_{14}$ | $(7,9,10,8,6,3,1,5)$ | $y_{14}$ | $(7,9,11,9,7,4,0,4)$ |
| $s_{15}$ | $(5,7,9,8,6,4,2,3)$ | $n_{15}$ | $(6,8,10,9,7,5,1,3)$ | $y_{15}$ | $(6,8,10,10,8,6,2,2)$ |
| $s_{16}$ | $(4,6,8,9,7,5,3,2)$ | $n_{16}$ | $(5,7,9,10,8,6,3,1)$ | $y_{16}$ | $(4,7,9,11,9,7,4,0)$ |

**Table 17. MCs for all the edges in $P_8^5$.**

| Edges $e$ | $\xi_m(e\|F_m)$ | Edges $e$ | $\xi_m(e\|F_m)$ | Edges $e$ | $\xi_m(e\|F_m)$ |
|---|---|---|---|---|---|
| $s_2s_1$ | $(2,4,6,8,8,6,4,2)$ | $s_1n_1$ | $(3,5,7,9,8,6,4,1)$ | $n_1y_1$ | $(2,6,8,10,9,7,5,1)$ |
| $s_3s_2$ | $(2,3,5,7,9,7,5,3)$ | $s_2n_2$ | $(1,4,6,8,9,7,5,3)$ | $n_2y_2$ | $(0,4,7,9,10,8,6,3)$ |
| $s_4s_3$ | $(2,2,4,6,8,8,6,4)$ | $s_3n_3$ | $(1,3,5,7,9,8,6,4)$ | $n_3y_3$ | $(1,2,6,8,10,9,7,5)$ |
| $s_5s_4$ | $(3,2,3,5,7,9,7,5)$ | $s_4n_4$ | $(3,1,4,6,8,9,7,5)$ | $n_4y_4$ | $(3,0,4,7,9,10,8,6)$ |
| $s_6s_5$ | $(4,2,2,4,6,8,8,6)$ | $s_5n_5$ | $(4,1,3,5,7,9,8,6)$ | $n_5y_5$ | $(5,1,2,6,8,10,9,7)$ |
| $s_7s_6$ | $(5,3,2,3,5,7,9,7)$ | $s_6n_6$ | $(5,3,1,4,6,8,9,7)$ | $n_6y_6$ | $(6,3,0,4,7,9,10,8)$ |
| $s_8s_7$ | $(6,4,2,2,4,6,8,8)$ | $s_7n_7$ | $(6,4,1,3,5,7,9,8)$ | $n_7y_7$ | $(7,5,1,2,6,8,10,9)$ |
| $s_9s_8$ | $(7,5,3,2,3,5,7,9)$ | $s_8n_8$ | $(7,5,3,1,4,6,8,9)$ | $n_8y_8$ | $(8,6,3,0,4,7,9,10)$ |
| $s_{10}s_9$ | $(8,6,4,2,2,4,6,8)$ | $s_9n_9$ | $(8,6,4,1,3,5,7,9)$ | $n_9y_9$ | $(9,7,5,1,2,6,8,10)$ |
| $s_{11}s_{10}$ | $(9,7,5,3,2,3,5,7)$ | $s_{10}n_{10}$ | $(9,7,5,3,1,4,6,8)$ | $n_{10}y_{10}$ | $(10,8,6,3,0,4,7,9)$ |
| $s_{12}s_{11}$ | $(8,8,6,4,2,2,4,6)$ | $s_{11}n_{11}$ | $(9,8,6,4,1,3,5,7)$ | $n_{11}y_{11}$ | $(10,9,7,5,1,2,6,8)$ |
| $s_{13}s_{12}$ | $(7,9,7,5,3,2,3,5)$ | $s_{12}n_{12}$ | $(8,9,7,5,3,1,4,6)$ | $n_{12}y_{12}$ | $(9,10,8,6,3,0,4,7)$ |
| $s_{14}s_{13}$ | $(6,8,8,6,4,2,2,4)$ | $s_{13}n_{13}$ | $(7,9,8,6,4,1,3,5)$ | $n_{13}y_{13}$ | $(8,10,9,7,5,1,2,6)$ |
| $s_{15}s_{14}$ | $(5,7,9,7,5,3,2,3)$ | $s_{14}n_{14}$ | $(6,8,9,7,5,3,1,4)$ | $n_{14}y_{14}$ | $(7,9,10,8,6,3,0,4)$ |
| $s_{16}s_{15}$ | $(4,6,8,8,6,4,2,2)$ | $s_{15}n_{15}$ | $(5,7,9,8,6,4,1,3)$ | $n_{15}y_{15}$ | $(6,8,10,9,7,5,1,2)$ |
| $s_1s_{16}$ | $(3,5,7,9,7,5,3,2)$ | $s_{16}n_{16}$ | $(4,6,8,9,7,5,3,1)$ | $n_{16}y_{16}$ | $(4,7,9,10,8,6,3,0)$ |
| $y_1n_2$ | $(1,5,7,9,10,8,6,2)$ | $y_7n_8$ | $(8,6,2,1,5,7,9,10)$ | $y_{12}n_{13}$ | $(8,10,9,7,4,0,3,6)$ |
| $y_2n_3$ | $(0,3,6,8,10,9,7,4)$ | $y_8n_9$ | $(9,7,4,0,3,6,8,10)$ | $y_{13}n_{14}$ | $(7,9,10,8,6,2,1,5)$ |
| $y_3n_4$ | $(2,1,5,7,9,10,8,6)$ | $y_9n_{10}$ | $(10,8,6,2,1,5,7,9)$ | $y_{14}n_{15}$ | $(6,8,10,9,7,4,0,3)$ |
| $y_4n_5$ | $(4,0,3,6,8,10,9,7)$ | $y_{10}n_{11}$ | $(10,9,7,4,0,3,6,8)$ | $y_{15}n_{16}$ | $(5,7,9,10,8,6,2,1)$ |
| $y_5n_6$ | $(6,2,1,5,7,9,10,8)$ | $y_{11}n_{12}$ | $(9,10,8,6,2,1,5,7)$ | $y_{16}n_1$ | $(3,6,8,10,9,7,4,0)$ |
| $y_6n_7$ | $(7,4,0,3,6,8,10,9)$ | | | | |

# Author contributions

**Conceptualization:** Sunny Kumar Sharma.

**Data curation:** Sunny Kumar Sharma.

**Formal analysis:** Sunny Kumar Sharma.

**Funding acquisition:** Vijay Kumar Bhat, Bandar Almohsen.

**Investigation:** Vijay Kumar Bhat.

**Methodology:** Vijay Kumar Bhat.

**Project administration:** Manikonda Gayathri.

**Resources:** Manikonda Gayathri.

**Software:** Manikonda Gayathri, Bandar Almohsen.

**Supervision:** Muhammad Azeem, Bandar Almohsen.

**Validation:** Muhammad Azeem.

**Visualization:** Muhammad Azeem.

**Writing – original draft:** Sunny Kumar Sharma, Vijay Kumar Bhat, Muhammad Azeem, Manikonda Gayathri.

**Writing – review & editing:** Sunny Kumar Sharma, Vijay Kumar Bhat, Muhammad Azeem, Manikonda Gayathri, Bandar Almohsen.

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
