## [Decision Letter · Decision Letter 0]

22 Aug 2024

PONE-D-24-20403Computation of mixed resolvability for a circular ladder consisting h-five sidesPLOS ONE

Dear Dr. Azeem,

Thank you for submitting your manuscript to PLOS ONE. After careful consideration, we feel that it has merit but does not fully meet PLOS ONE’s publication criteria as it currently stands. Therefore, we invite you to submit a revised version of the manuscript that addresses the points (appended below) raised during the review process.

Please submit your revised manuscript by Oct 06 2024 11:59PM. If you will need more time than this to complete your revisions, please reply to this message or contact the journal office at plosone@plos.org. Please include the following items when submitting your revised manuscript:

We look forward to receiving your revised manuscript.

Kind regards,

Rab Nawaz

Academic Editor

PLOS ONE

Journal Requirements:

Reviewers' comments:

Reviewer's Responses to Questions

**Comments to the Author**

1. Is the manuscript technically sound, and do the data support the conclusions?

Reviewer #1: Partly

Reviewer #2: Yes

2. Has the statistical analysis been performed appropriately and rigorously? 

Reviewer #1: N/A

Reviewer #2: N/A

3. Have the authors made all data underlying the findings in their manuscript fully available?

Reviewer #1: Yes

Reviewer #2: Yes

4. Is the manuscript presented in an intelligible fashion and written in standard English?

Reviewer #1: Yes

Reviewer #2: Yes

5. Review Comments to the Author

Reviewer #1: Referee’s Comments

Manuscript ID PONE-D-24-20403

The authors performed computation of mixed resolvability for a circular ladder consisting of h-five. The vertices and edges are identified using parameters within the domain of graph theory. They found that mixed metric dimension is unbounded, and it depends upon the number of vertices.

The comments are given

Don’t use subject pronoun in abstract.

Add more outcomes at the end of abstract.

In introduction author said with size k and order m and used \left|E\right|\ =\ k\ and\ \left|V\right|\ =\ m. so don’t mention these terms two times.

In introduction author should use number in ascending order for citation.

Author should write table 2 instead of table no. 2.

In reference list author should use same pattern for journal name.

Reviewer #2: The research article presents a significant contribution to the field of graph theory by focusing on the mixed metric dimension (MMD) of a unique planar graph family. The study thoroughly investigates the unbounded nature of the MMD in these circular pentagonal ladder graphs and demonstrates how it is dependent on the number of vertices. The results of this research not only provide a deeper understanding of the structural properties but also contribute valuable insights that can be applied to various scientific domains, including optimization, network design, and decision-making processes. One of the strengths of this paper is its rigorous mathematical analysis, which effectively bridges the gap between theoretical graph properties and potential practical applications. The authors carefully derive and analyze the mixed metric dimension, metric dimension, and edge metric dimension while drawing meaningful comparisons between them. This comparative analysis is insightful and highlights the rapid growth of the MMD relative to other resolvability parameters, which has important implications for understanding complex network structures. I have few observations to incorporate in the revised manuscript:

1. I recommend changing the title of the research article as "Computation of mixed resolvability for a circular pentagonal ladder its unbounded nature" making it more relevant and suitable in terms of conducted research.

2. The article lacks a clear motivation and fails to adequately explain the practical significance of studying this particular graph family. Although the study mentions applications in optimization, decision-making, and network design, it does not convincingly demonstrate how the theoretical findings translate to real-world problems. The discussion is largely confined to mathematical abstractions without connecting the results to concrete applications or providing examples of how the findings could be utilized in specific scenarios.

3. I would like to suggest replacing some of the older cited references with more recent and relevant studies. Incorporating these up-to-date references will enhance the credibility and relevance of your research. Below are the recommended citations:

Ali et al. (2024) IEEE Transactions on Industrial Electronics, 71(6), 6128-6138. doi: 10.1109/TIE.2023.3290247, Zheng et al. (2022) Robotics and Computer-Integrated Manufacturing, 73, 102238. doi: 10.1016/j.rcim.2021.102238, Guo and Wang, S. (2024) Communications in Algebra, 52(9), 3946-3959. doi: https://doi.org/10.1080/00927872.2024.2337276 and Qi et al. (2024) IEEE Transactions on Industrial Informatics, 20(4), 6631-6641. doi: 10.1109/TII.2024.3352232.

Please consider integrating these recent studies as replacements for some of the older references in your manuscript.

4. The presentation of the theoretical results is somewhat redundant and repetitive in few places. The paper spends more time reiterating known concepts in metric dimension theory which can be omitted thereby directly referring them to those known results.

In conclusion, this paper is a valuable addition to the existing literature on graph theory, particularly in the study of mixed metric dimensions in planar graphs. The research offers a fresh perspective on an interesting graph family and provides a strong theoretical foundation for future studies. The article is well-written, the results are novel, and the methodologies are sound, making it a suitable candidate for publication. Therefore, I support the publication of this article and recommend it for acceptance subject to addressing above minor points.

6. PLOS authors have the option to publish the peer review history of their article (what does this mean?). If published, this will include your full peer review and any attached files.

Reviewer #1: No

Reviewer #2: No

---

## [Author Response · Author response to Decision Letter 1]

14 Oct 2024

We have attached the file in response to the reviewer.

---

## [Decision Letter · Decision Letter 1]

31 Oct 2024

Computation of mixed resolvability for a circular ladder and its unbounded nature

PONE-D-24-20403R1

Dear Dr. Azeem,

We’re pleased to inform you that your manuscript has been judged scientifically suitable for publication and will be formally accepted for publication once it meets all outstanding technical requirements.

Kind regards,

Rab Nawaz

Academic Editor

PLOS ONE

Additional Editor Comments (optional):

Reviewers' comments:

Reviewer's Responses to Questions

**Comments to the Author**

1. If the authors have adequately addressed your comments raised in a previous round of review and you feel that this manuscript is now acceptable for publication, you may indicate that here to bypass the “Comments to the Author” section, enter your conflict of interest statement in the “Confidential to Editor” section, and submit your "Accept" recommendation.

Reviewer #1: All comments have been addressed

Reviewer #2: All comments have been addressed

2. Is the manuscript technically sound, and do the data support the conclusions?

Reviewer #1: Yes

Reviewer #2: Yes

3. Has the statistical analysis been performed appropriately and rigorously? 

Reviewer #1: Yes

Reviewer #2: N/A

4. Have the authors made all data underlying the findings in their manuscript fully available?

Reviewer #1: Yes

Reviewer #2: Yes

5. Is the manuscript presented in an intelligible fashion and written in standard English?

Reviewer #1: No

Reviewer #2: Yes

6. Review Comments to the Author

Reviewer #1: (No Response)

Reviewer #2: I endorse the response to comments highlighted during the review. Therefore, the revised manuscript can be accepted for publications.

7. PLOS authors have the option to publish the peer review history of their article (what does this mean?). If published, this will include your full peer review and any attached files.

Reviewer #1: No

Reviewer #2: No

---

## [Editor Report · Acceptance letter]

PONE-D-24-20403R1

PLOS ONE

Dear Dr. Azeem,

I'm pleased to inform you that your manuscript has been deemed suitable for publication in PLOS ONE. Congratulations! Your manuscript is now being handed over to our production team.

Kind regards,

on behalf of

Dr. Rab Nawaz

Academic Editor

PLOS ONE